



# Structural control on fluid flow and shallow diagenesis: Insights from calcite cementation along deformation bands in porous sandstones

Leonardo DEL SOLE[1], Marco ANTONELLINI[1], Roger SOLIVA[2], Gregory BALLAS[2], Fabrizio BALSAMO[3], and Giulio VIOLA[1]

[1]BiGeA - Department of Biological, Geological and Environmental Sciences, University of Bologna, Via Zamboni 67, 40126 Bologna, Italy

[2]Laboratoire Géosciences Montpellier, Université de Montpellier, CNRS, Université des Antilles, Montpellier, France.

[3]Next, Natural and Experimental Tectonic Research Group, Department of Chemistry, Life, Sciences and Environmental Sustainability, University of Parma, Parco Area delle Scienze 157A, 43124 Parma, Italy

*Correspondence to*: Leonardo Del Sole (leonardo.delsole@unibo.it)

**Abstract.** Porous sandstones are important reservoirs for geofluids. Interaction therein between deformation and cementation during diagenesis is critical since both processes can strongly reduce rock porosity and permeability, deteriorating reservoir quality. Deformation bands (DBs) and structural-related diagenetic bodies, here named Structural and Diagenetic Heterogeneities (SDH), have been recognized to negatively affect fluid flow at a range of scales and potentially lead to reservoir compartmentalization, influencing flow buffering and sealing during production. The hydraulic behavior of DBs is not yet fully constrained, and it remains poorly understood also how diagenetic processes interact with DBs to steer fluid flow mechanisms and evolution. In this contribution we present two field-based studies from Loiano (Northern Apennines, Italy) and Bollène (Provence, France) that contribute to elucidating the structural control exerted by DBs on fluid flow and diagenesis recorded by calcite nodules associated with the bands. We relied on careful field observations and a variety of multiscalar mapping techniques (photography, string mapping, and drone aerial photography), integrated with optical, scanning electron and cathodoluminescence microscopy, and stable isotope ($\delta^{13}C$ and $\delta^{18}O$) analysis of nodules cement. In both case studies, at least one set of DBs precedes and controls selective cement precipitation. Cement texture and cathodoluminescence patterns, and their invariably negative $\delta^{13}C$ and $\delta^{18}O$ value ranges, suggest a meteoric environment for nodule formation. In Loiano, DBs acted as low-permeability barriers to fluid flow and promoted selective cement precipitation. In Bollène, clusters of DBs restricted fluid flow and focused diagenesis in parallel-to-band compartments. Our work shows how low-permeability DBs in porous sandstones can actually affect fluid flow and localize diagenetic processes (in the shallow crust) that, in turn, could further enhance the sealing capacity of these structural features.



# 1 Introduction

Porous rocks, such as sandstones and carbonates, are important reservoirs for geofluids. In this kind of granular materials, structural and diagenetic processes commonly affect the petrophysical properties and reservoir quality. The importance of the interaction between deformation and deformational structures, fluid flow, and diagenetic processes has been emphasized only during the last two decades (e.g. see the recently coined term "*structural diagenesis*", Laubach et al., 2010; Mozley and Goodwin, 1995; Eichhubl et al., 2009; Balsamo et al., 2012; Philit et al., 2015; Antonellini et al., 2017,

2020; Del Sole et al., 2020). If deformation influences diagenesis and vice-versa, a feedback can eventually develop between these two processes. For example, early diagenesis can influence the mechanical properties of rocks (Antonellini et al., 2020), and, in turn, their mechanical stratigraphy (Laubach et al., 2009; La Bruna et al., 2020). Additionally, also structural and diagenetic heterogeneities (referred to as SDH from now on) can determine the texture as well as the petrophysical and mechanical properties of the rock volume hosting them (Antonellini and Aydin, 1994; Aydin, 2000; Faulkner et al., 2010;

Bense et al., 2013; Pei et al., 2015; Del Sole et al., 2020). Cement precipitation in granular porous siliciclastic rocks leads to porosity loss and reduction in permeability (Tenthorey et al., 1998; Morad et al., 2010) and, in turn, overall reservoir quality deterioration (Ehrenberg, 1990; Morad et al., 2010). Carbonate cement is commonly concentrated within a few, specific horizons or nodules with various shapes and arrangements (Kantorowicz et al., 1987; Bjørkum and Walderhaug, 1990; Mozley and Davis, 1996) making porosity and permeability prediction more complex (Davis et al., 2006; Morad et al.,

2010). Furthermore, cement increases the mechanical strength of the host rock (Dvorkin et al., 1991; Bernabé et al., 1992, Boutt et al., 2014) influencing fault-zone architecture and potential fault reactivation (Dewhurst and Jones, 2003; Flodin et al., 2003; Wilson et al., 2006; Williams et al., 2017; Philit et al., 2019; Pizzati et al., 2019).

Granular or porous sediments and sedimentary rocks commonly contain sub-seismic resolution strain localization features referred to as "deformation bands" (Aydin, 1978; DBs from now on). Fluid-flow mechanisms and evolution within

DBs, remain also poorly understood particularly with regards to diagenetic processes. The effect of DBs on fluid flow can significantly vary, with them acting as, e.g., conduits, baffles or barriers. Additionally, DBs can introduce a permeability anisotropy and compartmentalize reservoirs, and cause poor well performance. This might impact negatively upon production from faulted siliciclastic systems (Edwards et al., 1993; Lewis and Couples, 1993; Leveille et al., 1997; Antonellini et al., 1999; Wilkins et al., 2019) and flow-based models and simulations (Sternlof et al. 2004; Rotevatn and

Fossen, 2011; Fachri et al., 2013; Qu and Tveranger, 2016; Romano et al., 2020). In some cases, DBs appear to be even conduits for fluids (e.g. Parry et al. 2004; Sample et al. 2006; Busch et al., 2017). Their overall different behavior depends on several factors, such as their permeability contrast relative to the host rock, their thickness, density, distribution (clustering), orientation, segmentation, and connectivity (Antonellini and Aydin, 1994; Gibson, 1998; Manzocchi et al., 1998; Sternlof et al., 2004; Shipton et al., 2005; Fossen and Bale, 2007; Torabi and Fossen 2009; Rotevatn et al., 2013). In turn, the factors are

controlled by the tectonic regime, burial conditions, and host rock characteristics (porosity, grain-size, grain-sorting, clay content; Antonellini et al., 1994, 1999; Antonellini and Mollema 2002; Balsamo and Storti, 2010; Soliva et al., 2013, 2016;



Ballas et al., 2014, 2015; Fossen et al., 2017; Philit et al., 2018). Cement precipitation in DBs may significantly increase the reduction of porosity and permeability caused by mechanical crushing and reorganization of grains, thus increasing their sealing or buffering potential (Edwards et al., 1993; Leveille et al., 1997; Fisher and Knipe, 1998; Parnell et al., 2004).

Different processes account for the presence of DBs and the occurrence of cement. For instance, in arid to semiarid vadose zones, DBs would enhance unsaturated flow relative to the host rock (Sigda et al., 1999; Sigda and Wilson, 2003) that could be gravity driven (Parry et al., 2004; Wilson et al., 2006) or the result of capillary suction along the fine-grained material in the band (Cavailhes et al., 2009; Balsamo et al., 2012). A transient increase in permeability associated with initial dilation in the early stage of DB formation (e.g. Antonellini et al., 1994; Main et al., 2000) may focus fluid flow and other processes

(e.g. cementation, hydrocarbon inclusion entrapment, removal of iron oxide coatings) in and around the band (Fowles and Burley, 1994; Labaume and Moretti, 2001; Ogilvie and Glover, 2001; Parnell et al., 2004; Parry et al., 2004; Sample et al., 2006; Lommatzsch et al., 2015). In the aforementioned processes, however, it is implicit that DBs behaved as "conduits" in order to explain the occurrence of cement or other authigenic products within these structures. Nevertheless, a significant number of studies on DBs show that in most of cases they are baffle or seals to fluid flow (see Ballas et al., 2015 for a

review). Much less work has been addressed to elucidate the fluid flow and diagenetic mechanisms leading to selective cementation in association with low-permeability baffle/barrier DBs (Philit et al., 2015).

A cross-disciplinary study integrating structural (DB characteristics) and diagenetic (cement characteristics) analysis is thus required to reliably constrain the combined structural and diagenetic evolution of the studied rock volumes (Laubach et al., 2010). The objective of this work is to investigate the interaction between DBs, fluid flow, and diagenesis. In

particular, our aim is to elucidate the hydraulic behavior of DBs in porous sandstones as well as fluid flow mechanisms and evolution, with regards to the diagenetic processes, related to such structures. We do this by examining two different field sites in Italy and France where calcite cement nodules are clearly spatially associated with DBs. The comparison between two locations with different geological settings makes it possible to derive conclusions of general validity that can be extended to other cases where DBs and diagenetic processes interact. Our study allows also to evaluate the impact of both

structural and structural-related diagenetic heterogeneities (SDH) on present-time fluid circulation (hydrological implications) and on subsequent deformation (mechanical implications).

## 2 Geological framework

### 2.1 Loiano field site, Northern Apennines (Italy)

The site of Loiano is in the Northern Apennines (Emilia-Romagna region, Italy), 20 kilometers to the south of the

city of Bologna (Fig. 1a). The Northern Apennines are an orogenic wedge formed in response to the Upper Cretaceous-Eocene closure of the Ligurian-Piedmont ocean (Marroni et al., 2017) and the subsequent Oligocene-Miocene convergence and collision between the Adriatic Promontory and the Sardinia-Corsica Block, of African and European origin, respectively (Vai and Martini, 2001). Our work focused on the Loiano Sandstones of the Epiligurian Successions (Fig. 1a-c), the Middle





Eocene to Middle Miocene siliciclastic infill of thrust-top, piggy-back basins discordant to the underlying Ligurian units,

which migrated passively to the NE during the Apennines orogeny atop of the entre orogenic wedge. The 300-1000 m thick, Late Lutetian-Bartonian Loiano Sandstones are a fan-delta to proximal turbidite deposit (Papani, 1998). They are medium- to coarse-grained, poorly consolidated, immature arkosic sandstones and conglomerates deposited in a relatively small lenticular basin (a few tens of km in width and length; Fig. 1a, c). They are composed of 49 to 60% quartz and of 39–48% feldspar, the rest being rock fragments, detrital carbonate clasts, and minor accessories (Del Sole and Antonellini, 2019).

**2.2 Bollène field site, Southeast Basin (Provence, France)**

The site of Bollène is in the Southeast Basin of Provence (France), 15 kilometers to the north of the city of Orange (Fig. 2a). The Southeast Basin is a triangular region between the Massif Central to the north-west, the Alps to the east, and the Mediterranean Sea to the south. It is a Mesozoic cratonic basin on the edge of the Alpine orogen, approximately 200 km long and 100 to 150 km wide. Three main tectonic episodes affected the region (Arthaud and Séguret, 1981; Roure et al.,

1992; Séranne et al., 1995; Champion et al., 2000): SSW-NNE Pyrenean contraction from Paleocene to Oligocene, NW-SE Gulf of Lion extension from the Oligocene to early Miocene (rifting), and, lastly, SW-NE Alpine contraction from Miocene to Quaternary (Fig. 2a). The site of Bollène is exposed in a quarry (Figs. 2c) located in Turonian sand (low cohesion sandstone), between 10 and 200 m thick in thickness and is situated north of the E-W Mondragon anticline (Fig. 2b, c). The Turonian sands at the Bollène quarry are laminated, fine to coarse grained with modal and bimodal grain size distributions;

they formed in deltaic and aeolian environments. The host sands are not cemented. They are composed of 88 to 92% quartz, the rest being feldspar. The median grain diameter ($D_{50}$) is 0.31 mm, i.e. medium sand. Their porosities range from 20 to 43%, and the precise value at study site is 22.05% (Ballas et al., 2014).







**Figure 1.** (a) Schematic geologic map and (b) cross-section of the Northern Apennines near Bologna (Italy), modified from Picotti and Pazzaglia, 2008. (c) Geologic map of the study area and location of the studied outcrops (red dotted line). This map is constructed from data of the *Regione Emilia-Romagna* (http://www.regione.emilia-romagna.it/). Location of (c) is indicated by a red square in (a). (d) Lower-hemisphere equal-area projection indicates the orientation of the different sets of DBs (298 data points) and poles to bedding at the study site. DBs associated with carbonate nodules are highlighted by a red line. DBs azimuth (± 90°) and dip angle (°) plotted against
frequency. Best-fit Gaussian curves superimposed on the corresponding data histograms (frequency distributions). Gaussian peaks, and related standard deviations (± sd), are also indicated.









**Figure 2.** (a) Schematic geologic map and (b) cross-section of the South East Basin, Provence, France. The main tectonic episodes
affecting the region are reported in (a). (c) Geologic map and stratigraphic column of the Bollène quarry. Location of map in (c) is
indicated by a red square in (a), and red open circles indicate some of past studies locations (see Wibberley et al., 2007; Saillet and
Wibberley, 2010; Ballas et al., 2012, 2013, 2014; Soliva et al., 2013; Philit et al., 2018). (a), (b), and stratigraphic column in (c) are
modified from Philit et al., (2015, 2018). Geological map in (c) is modified from Ballas et al. (2012). (d) Lower-hemisphere equal-area
projection indicates the orientation of the different sets of DBs (64 data points) at the study site. DBs associated with carbonate nodules are
highlighted in red. Dotted lines indicate the main attitude of tabular carbonate nodules. DBs azimuth (± 90°) and dip angle (°) plotted
against frequency. Best-fit Gaussian curves superimposed on the corresponding data histograms (frequency distributions). Gaussian peaks,
and related standard deviations (± sd), are also indicated. Number in square brackets [n] are the same as used in Fig. 6b to rank different
sets of DBs.

## 3 Methods

### 3.1 Outcrops analysis

The geometry and distribution of DBs and nodules were documented by detailed field mapping at different scales
for both sites. At the Loiano site, a map (370 m$^2$) at the 1:25 scale (1 cm ≃ 4 m) was made by standard topographic compass
and tape mapping (Fig. 3). The Bollène quarry site pavement was mapped using a DJI PHANTOM™ drone. Photographs
were taken at different altitude above the ground surface and were then used to build a 3D mesh and extract high-resolution
orthophotos using *Agisoft PhotoScan Metashape* software (© Agisoft LLC). The high-resolution orthophoto mosaic (1 px ≃
1-1.5 mm) was used for the detailed mapping of DBs and nodules. Furthermore, DBs and nodule patterns, as well as their
characteristics and spatial relationships, were documented in the field on high-resolution photographs (15 megapixels), both
in Loiano (Figs. 4 and 5) and Bollène (Figs. 7 and 8). Oriented samples were collected for thin section preparation,
microstructural, and stable isotopes analysis. The orientation and dip of DBs were measured at each site and plotted in lower
hemisphere equal area stereograms and frequency histograms (Figs. 1d and 2d) using the *Daisy3* software (Salvini, 2004).

### 3.2 Microstructural analysis

Polished thin sections of host sandstones, DBs, and nodule samples were analyzed by natural-light microscopy, cold
cathodoluminescence, and backscattered electron imagery using a JEOL JSM-5400, and a FEI Quanta FEG 200
environmental scanning-electron microscope (SEM). These microscopy techniques were used to examine the texture and
microstructures of host rock and DBs, as well as the cement distribution and texture (Figs. 9, 10, and 11). In particular, cold
cathodoluminescence (CL) analysis of carbonate cement in nodules was conducted with a CITL Cold cathodoluminescence
8200 Mk5-1 system (operated at 14-15kV beam energy and 250μA beam current) equipped with a standard petrographic
microscope (Olympus BH41). CL was used to describe the cement-crystals properties (texture, fabric, luminescence) and the
micron-scale spatial distribution and textural relationship among the cements, framework detrital grains, and the fractures



(*sensu lato*). This information could be used to (i) understand the interrelation between deformation, fluid flow, and diagenesis (e.g. cement precipitation); (ii) assess the relative timing of each process; (iii) describe porosity evolution with time; and (iv) understand the mechanisms and the geochemical environment of cement precipitation when coupled with other tools (e.g. stable isotopes analyses).

### 3.3 Stable isotope characterization

Stable carbon and oxygen isotope data from cements from within carbonate nodules were used to constrain the geochemical environment of precipitation and possible source of fluids. Powder samples for bulk rock carbon and oxygen stable isotopes analysis were ground with a dental drill from not weathered or altered sections of the nodules. A total of 46 sites were sampled from nodules in Loiano (n=30; Fig. 12a) and Bollène (n=16; Fig. 12b). Powders samples were analyzed with a Thermo Finnigan DELTA plus XP mass spectrometer coupled with a Thermo Finnigan Gas Bench II gas preparation
and introduction system. $\delta^{13}C$ and $\delta^{18}O$ are referred to the international standard V-PDB (Vienna-Pee Dee Belemnite). Isotope determination analytical precision was 0.10‰ and 0.15‰ V-PDB for carbon and oxygen, respectively. The prediction uncertainty was c. 0.15‰ for carbon and c. 0.20‰ for the oxygen isotopes.

### 4 Deformation bands and cement: field observations

#### 4.1 Loiano

At the study site, bedding is oriented NW-SE and dips to the NE with an average angle of 38° (Fig. 1d). Deformation bands occur in three different trends striking NNW-SSE, NNE-SSW to NE-SW, and ENE-WSW (Fig. 1d). All DBs dip moderately to steeply mostly in the west and south quadrants (Fig. 1d). Deformation bands commonly occur with a positive relief and appear as whitish linear traces with minor undulations forming eye and ramp structures, where they branch and merge (Figs. 3 and 4). Already at the outcrop they exhibit a significant reduction in grain size and porosity in
comparison to the surrounding host rocks (Fig. 4f). Deformation bands occur both as single features and as clusters or zone of bands, i.e. in narrow zones with variable thickness (0.8-60 cm) with sub-parallel DBs (up to 40). Single DBs accommodate minor offsets from a few millimeters up to 40 mm, whereas clusters can accommodate offsets up to 0.5 m (Fig. 4b, d). Deformation bands display a variety of apparent normal and strike-slip offsets (Fig. 3). Different sets of DBs show ambiguous and conflicting crosscutting relationships. Field observations indicate that the NNW-SSE and NNE-SSW
sets have mutual crosscutting relationships typical of faults forming synchronously (Fig. 3). They occur along all the exposed sandstone sequence but display different patterns wherein the grain size, sorting or porosity differ. In poorly sorted and/or low porosity coarse-grained sandstone, deformation appears more localized, and the trace of the DBs tends to be straight and develop along a single strand (Fig. 5e). In well-sorted and/or high porosity fine-grained sandstone, deformation





is more distributed and the trace of the DBs splits into several wavy and anastomosing segments (Fig. 5e). A clay smear
occurs where DBs cut through thin, dark-colored clay levels (Fig. 5e).

The peculiar characteristic of the Loiano Sandstones is the presence of spatially heterogeneous carbonate cementation. It has led to isolated or multiple spheroids or irregularly shaped nodules and continuous tabular nodules (Figs. 4 and 5). The nodules weather out in positive relief, because they are more resistant to weathering than the weakly cemented host rock. Isolated nodules range in diameter (major horizontal axis) from 0.2 m to 3 m (Fig. 4c, d) whereas tabular
concretions have a thickness ranging from 0.10 to 0.8 m and a long axis ranging from 3 up to 15 m in length (Fig. 5a). Generally, the nodule shape in Loiano is similar to that of an oblate spheroid, since two of the three spheroid dimensions (i.e. length and width) are greater than the third (i.e. thickness). There is no evidence of spherical nodules or prolate spheroids. The volume of the carbonate nodules ranges from 0.001 $m^3$ to > 10 $m^3$. Nodules form about 20% of the exposed outcrop volume. Two types of nodules can be distinguished depending on whether they are associated with DBs or clusters (i.e. DBs-
parallel nodules; Figs. 3, 4, and 5a-c) or with bedding planes (i.e. bedding-parallel nodules; Figs. 3 and 5d, e). The former represent roughly 75% of the total nodules in the study area and are the main target of this work. The association between DBs and nodules occurs in the form of (i) parallelism and spatial overlap between DBs and nodules and (ii) confinement of the nodules by the DBs. In all cases, nodules are oriented with the major axis (elongation direction) parallel to the DBs and the minor-axis (i.e. thickness) perpendicular to them. Deformation band-parallel nodules are isolated ellipsoids (Fig. 4b, d),
or, alternatively, continuous tabular objects (Figs. 4a and 5a). Nodules may be located along the DB (or zone of bands) trace (Figs. 4a, d and 5a, b), placed in between and confined by DBs (Figs. 4b, e, and 5c), or they may be asymmetrically placed on one side of the DBs (Figs. 4c and 9e). In some cases, nodules lie at the intersection of different DBs planes (Fig. 3). Some DBs are not spatially associated with nodules (Figs. 3 and 4b). Among the multiple sets of DBs, those mostly associated with carbonate nodules are the NNW-SSE and the NNE-SSW one. As a result, most nodules are elongated along these two
structural directions (Figs. 1d and 3; see also Fig. 1b in Del Sole and Antonellini, 2019). The other sets are rarely associated with carbonate nodules. Nodules are never cut across by the DBs.



**Figure 3.** Outcrop map that documents geometry and distribution of DBs and nodules in a portion of the study area in Loiano. The right-hand panel fits on top of the left-hand panel. ZB – zone of bands. The inset (© Google Earth) shows the map location in the study area.



Bedding-parallel nodules occur with different geometries: isolated (Fig. 3), multiple but laterally discontinuous (Figs. 3 and 5d, e), or laterally continuous layers with a tabular geometry (Fig. 3; e.g. "nodular beds" in Del Sole et al., 2020). Nodular beds are continuous pervasively-cemented layers that extend along the bedding plane for several meters (up

to 15 m in length) and a nearly constant thickness of c. 35-50 cm. Nodules along bedding planes are more rounded and with gentle boundaries (Fig. 5d, e) than those associated with DBs, which are more tabular and exhibit angular and sharp boundaries (Figs. 4 and 5a-c). In some cases, nodule geometry and elongation direction follow both bedding surfaces and DBs (Fig. 3). Nodules, despite being ubiquitous in the sandstone, are mostly observed within coarse levels with grain size equal or larger than medium sands (0.25-0.5 mm). We did not observe any nodules in sedimentary rocks with grain size finer

than sand (siltstone and clay). Bedding-parallel nodules are commonly located in sandstone levels confined between clay/silty levels or fine-grained-sand levels (Figs. 3 and 5d, e).

A set of joints and veins (Fig. 5a, b) were found exclusively within the carbonate nodules. They postdate DBs and nodules and do not propagate into the surrounding host sandstone.





**Figure 4.** Relationships between nodules and DBs. Deformation bands occur either as single structures or organized in clusters (ZB). Nodules along DBs (or ZB) are isolated (b, d, e), or continuous with a tabular geometry (a). Nodules are located along the DB trace (a, b, d), or they are asymmetrically placed on a side of the DB (c). (e) Isolated nodule placed in between and confined by ZB. DBs exhibit a whitish color with respect to the host rock, a positive relief, and a clear reduction in grain size and a lower porosity, both visible to the naked eye (f). The pen in (c, d) is 14 cm in length. The arrow-scale in (b, e) is 10 cm in length. The position of (a, b) is indicated in Fig. 3.





**Figure 5.** (a-c) DB-parallel- and (d-e) bedding-parallel nodules. (a) Decametric-scale continuous nodule with a tabular geometry located along a zone of bands (ZB). NE-dipping layering is shown with dotted lines. The photo is about 10 meters in depth. (b) Close-up on (a) from a map-view. The lens-cover is 5.5 cm in diameter. (a, b) Late opening-fractures cut through the assemblage "DBs – nodule" and they do not propagate into the poorly consolidated host rock. (c) Isolated nodules placed in between and confined by ZB. (d, e) The bedding is



emphasized mostly by clay and silt levels, sporadic well-defined thin levels of gravel, and the alignment of bedding-parallel nodules. (d) Black arrows point to multiple but laterally discontinuous bedding-parallel nodules. Here, a single DB (white line) crosses a bedding-parallel nodule without causing any offset. The position of (d) is indicated in Fig. 3. (e) Photomosaic showing a series of laterally-

discontinuous nodules just below, above or in between several continuous impermeable clay-rich levels. The deformation pattern changes depending on the host rock properties (e.g. sorting degree, porosity $\phi$, grain-size). Cataclastic deformation is accompanied by clay smear (see inset) where DBs cut thin dark-colored clay levels.

## 4.2 Bollène

At Bollène, DBs occurs as belonging to three different trends oriented (i) NW-SE to NNW-SSE (set 1 in Figs. 2d

and 11b), (ii) NE-SW to ENE-WSW (sets 2 and 3 in Figs. 2d and 11b), and (iii) ESE-WNW (set 4 in Figs. 2d and 11b) orientation. Trend (i) can be divided in two subsets; one is characterized by normal offsets NW-SE conjugate bands, moderate dip angles (50-60°) to SW (Figs.7a and 8a, c), and just a few to NE; a second one is characterized by dominant dextral strike-slip kinematic bands with higher dip angles (70-90°), and a NNW-SSE trend (Figs. 6b and 7f). Trend (ii) can be divided in two sets; one set is characterized by dominantly left-lateral and minor right-lateral subvertical strike-slip

conjugate bands striking NE-SW to ENE-WSW (set 2 in Fig. 6b; Fig. 7a-c); a second set is instead characterized by a set of conjugate DBs with moderate dips (~60°), NE-SW orientation (set 3 in Fig. 6b), and undetermined kinematic (likely normal-sense). Trend (iii) is composed of ESE-WNW conjugate bands, with reverse kinematics and low dip angles (30-40°). In the field, DBs appear as whitish linear traces with minor undulations and characteristic eye structures where they branch and merge (Figs. 6b and 7). In most cases, DBs weather out in positive relief. Frequently DBs occur in narrow zones (a few

millimeters up to 5-15 cm in thickness). Field observations indicate that ESE-WNW DBs are crosscut by NE-SW strike-slip DBs (Fig. 6b). The latter also crosscut also the NW-SE/NNW-SSE set (Fig. 7a-c, f). Bedding is oriented NW-SE and dips gently (<10°) to the S (Fig. 2b, d). Bedding is difficult to recognize on the floor of the quarry, because of its low dip and the massive texture of the rock (Fig. 6b). The Turonian sandstones outcrop on the quarry floor. There are two lithotypes. The first one is represented by massive porous sands with DBs and localized carbonate cementation (see description below). The

second one is characterized by a massive calcrete level with tabular geometry (see Supplement for details).

The Turonian Sandstones in the Bollène quarry are characterized by a spatially heterogeneous cementation (Fig. 6b). These diagenetic heterogeneities occur as spherical and tabular nodules (Figs. 7 and 8). Spherical nodules are arranged as isolated bodies within the surrounding host rock (Figs. 7c, g and 8a, d, e) or aggregated in tabular clusters (Fig. 7a, d). Nodules weather out in positive relief. Spherical nodules range in diameter from a few millimeters (0.004-0.005 m) to a few

tens of cms (0.2 m), whereas tabular ones have a thickness ranging from a few cms to 0.1 m and a long axis up to 5 m in length (Figs. 6b and 7a-c). Assessment of nodules lateral extension is hampered by the presence of vegetation and debris cover whereas subsurface extension cannot be measured, because of the limited vertical exposures of the outcrops. Hence, the values reported here are minimum values. In general, the nodule shape may be approximated by a sphere where length, width, and thickness are "equal" and by an oblate spheroid where length and width are larger than the nodule thickness.



Carbonate nodule volume ranges from $10^{-7}$ (small spherical nodules) to $> 2.5$ m$^3$ (tabular nodules assuming length = width = 5m, and thickness = 0.1m). In Bollène, the nodules are all spatially and geometrically associated with DBs (i.e. DBs-parallel nodules). This association occurs in the form of (1) parallelism between DBs and nodules, (2) geometric congruence between the DBs trend and the nodule (or nodules cluster) shape, and (3) confinement of nodules in parallel-to-bands compartments. In all these cases, tabular nodules and clusters of spherical nodules are oriented with the major axis

(elongation direction) parallel to the DBs and the minor-axis (i.e. thickness) perpendicular to them (Figs. 7a and 8a). Unlike what we have seen in Loiano, in Bollène nodules are in compartments among DBs. Carbonate nodules are associated with the NW-SE/NNW-SSE DBs set (Figs. 6b and 7a, f). Although this set is conjugate with bands dipping to the SW and to the NE, the tabular cement bodies dip only to the SW (Figs. 2d and 8a-c). No nodules are cut by the NW-SE bands. The NE-SW/ENE-WSW strike-slip bands cut through the NW-SE/NNW-SSE bands and the associated NW-SE-trending carbonate

nodules (Figs. 7a-c, f). There is clear evidence of these crosscutting relationships both at the outcrop and at the micro-scale (see Sect. 5.2). For this reason, we focus on the NW-SE DBs and nodules in the remaining part of this study.





**Figure 6.** (a) Aerial photograph of the study site (© Google Earth). (b) Orthophoto that documents geometry and distribution of DBs and carbonate nodules in the Bollène quarry. Lower-hemisphere equal-area projection indicating the orientation of the cataclastic structures measured in (1) NW-SE/NNW-SSE normal and (dextral) strike-slip bands associated with tabular and spherical nodules; (2) NE-SW/ENE-WSW strike-slip bands; (3) NE-SW bands with undetermined kinematics; (4) ESE-WNW reverse-sense bands.







**Figure 7.** Calcite cement occurs in the form of isolated (a, c, g) or clusters (a, d) of (S) spherical nodules and continuous (t) tabular nodules (a-c). Nodules are arranged in compartments parallel to clusters of NW-SE normal bands (a-c) and to NNW-SSE dextral strike-slip bands (f, g). NE-SW/ENE-WSW strike-slip bands displace both the NW-SE bands and their associated nodules (a-c, f). (e) Spherical nodule (about 5 cm in diameter) in spatial superposition with a NW-SE DB that does not displace the cement. Lens-cover in (a, f) has a 5.5 cm diameter. The figures (a-d, f, g) are in map-view. The position of (a, f) is indicated in Fig. 6b. Hammer length in (b) is 30.5 cm.



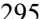


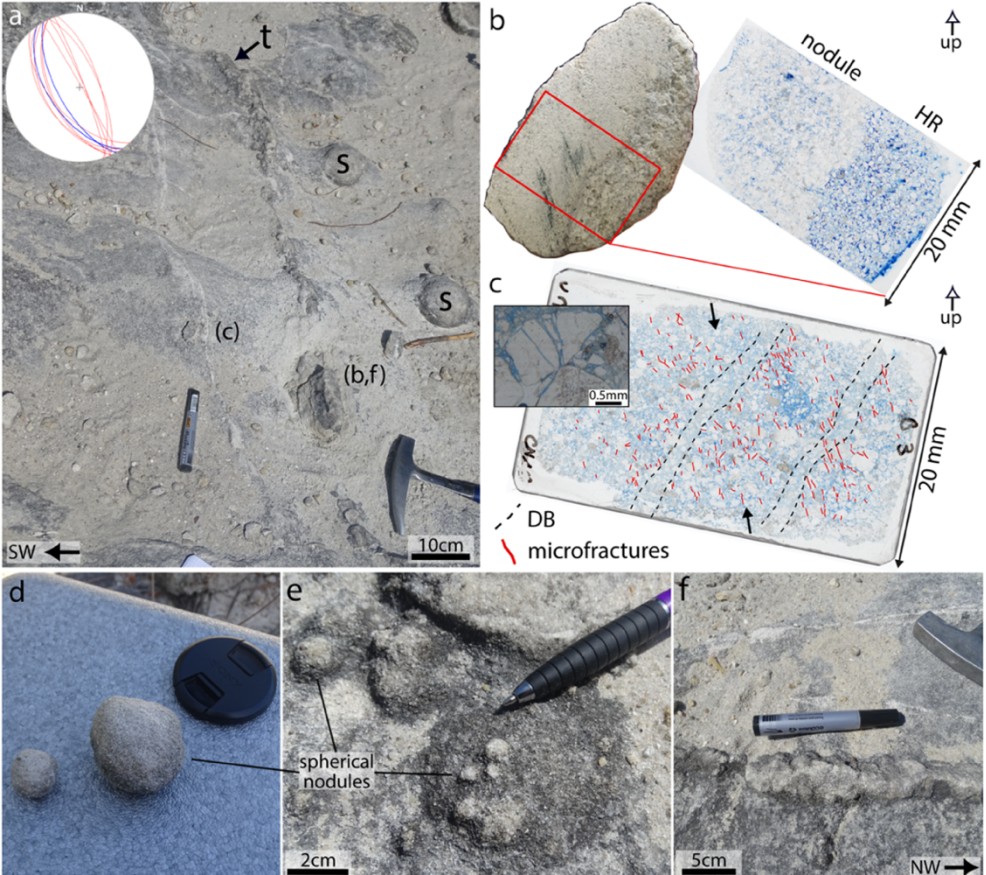

**Figure 8.** Typical relationships between nodules and DBs. (a) Continuous tabular nodule (t) and isolated spherical nodules (S) aligned parallel to the NW-SE normal-sense bands dipping SW (~55°; red lines in the inset stereoplot). The tabular bodies dip to the SW parallel to the bands (blue lines in the stereoplot in the inset). The position of (a) is indicated in Fig. 6b. (b) Hand specimen and polished thin sections impregnated with blue-dyed resin of a tabular nodule in (a). (c) Polished thin sections impregnated with blue-dyed showing two sub-parallel NW-SE normal-sense bands dipping SW. Mapping of microfractures developed at grain contacts consistent with Hertzian contacts (e.g. Eichhubl et al., 2010; Soliva et al., 2013) endorses the normal kinematic of these bands. "Up" refers to the topography. Close-up on spherical (d-e) and tabular (f) nodules from the field. Pen-marker length in (a, f) is about 13.5 cm. Lens-cover in (d) has a 5.5 cm diameter.

## 5 Deformation bands and cement: textural and microstructural characteristics

### 5.1 Loiano

Figure 9 reports data of host rock and DBs porosity in the Loiano Sandstones. Host rock total porosity (minus-cement $\phi$) is between 20-26%. Porosity is predominantly intergranular, whereas intragranular (e.g. pores within bioclasts)



and "oversize" pores are due to dissolution of detrital grains (Figs. 9 and 10). Deformation band total porosity (minus-

cement $\phi$) is lower by an order of magnitude (below 5%) than the host rock porosity (Fig. 9d). In the nodules, the host rock porosity is almost completely filled by cement (Fig. 10), so that the remnant porosity (voids) is low (down to 1.3%) (Fig. 9a, c). The presence of cement within the DBs enhances the porosity reduction caused by grain crushing and compaction (Fig. 9d).

        The microstructure of the DBs is characterized by reduced of grain size, porosity and pore size than in the host rock

(Fig. 10i-l). Within the DBs a few coarse grains are surrounded by a fine-grained matrix. Grains in the DBs are fractured and characterized by angular shapes, whereas the host rock consists of more rounded and nearly undisturbed medium- to coarse-grained sand grains (Figs. 9b-d and 10). However, some intragranular fractures are present at grain-to-grain contacts (Fig. 10a-b). These microfractures are common in the boundary zone of the DBs (Fig. 10i). Microfractures in larger grains are often filled by fine-grained clasts. Detrital calcite clasts and bioclasts are often fractured or affected by pressure solution at

contact points with other framework grains (Fig. 10b-d). Elongated detrital mica grains (mostly biotite) are bent between other grains and they tend to be preferentially aligned parallel to the bedding (Fig. 10c). In some cases, where framework grains are in contact, they show straight-elongate planar to slightly undulated grain to grain contacts (Fig. 10b) that are also roughly parallel to the bedding. Locally, we observed partial dissolution of feldspar with formation of authigenic calcite (Fig. 10e). Minor calcite cement occurs also as syntaxial overgrowths around fossil shells and detrital carbonate grains (Fig. 10f).

Despite these different effects of mechanical and chemical compaction and minor authigenic alterations, the major diagenetic components of the Loiano Sandstone are calcite cements. These cements fill mainly intergranular, and at lesser extent intragranular (intraskeletal) pore spaces, intragranular fractures, and they encase the framework grains and all other diagenetic features described above. Bedding-parallel nodules (Fig. 10a-h) are characterized by a mosaic texture of blocky sparite to poikilotopic bright-orange to orange-CL calcite cement. Crystal-size is typically 40-100 μm and up to 300 μm (Fig.

10g, h). This cement phase is the most widespread one and it is almost everywhere uniform in terms of texture and CL-pattern, if not for some minor dark-CL sub-zones (Fig. 10g, h). A minor calcite cement phase is associated with detrital carbonates (bioclasts) and it shows a bright-orange CL (Fig. 10c-d). It occurs as pore-lining formed by elongate and sharp rhombohedral shape calcite (dogtooth) that outlines the outer rim of the bioclasts (Fig. 10c) and drusy mosaic calcite that fills intraskeletal pores, outline bioclasts, and fills intragranular fractures in bioclasts (Fig. 10d). This subordinate phase was

observed only in bedding-parallel nodules. Pore-lining cement around bioclasts is present only where there was pore-space (now filled), whereas it is absent where other grains are in contact with the bioclast. All cement phases described above (intergranular, intraskeletal, and pore-lining) encase compacted grains, and cements are undeformed still preserving the original shape. Pore-filling cement in DBs-parallel nodules (Fig. 10i-n) show a similar texture and CL-pattern to that described for the main intragranular cement phase in the bedding-parallel nodules (Fig. 10a-h). The main features that

differentiate DB-parallel nodules from bedding-parallel ones are the finer crystal size of calcite within DBs, (Fig. 10i-l) and the absence of a bright-orange CL cement phase described in association to bioclasts in bedding-parallel nodules (Fig. 10c, d). Some bright-orange CL cement was observed only in detrital form (crushed) within the DB (Fig. 10k). The pore filling in





DBs is fine-grained sparite; no evidences were found of crushed calcite crystals belonging to the dominant calcite phase. The finer fraction within the DBs is a matrix (flakes) made up of comminuted angular and fine-grained clasts of feldspar and in a

smaller degree quartz, encased by the cement (Fig. 10j-l). Similarly, the cement fills the microfractures that cut through coarser grains and it encase the fine-grained clasts that are present within these fractures. Although, these microfractures are frequent in the host rock sectors in proximity to the DB (Fig. 10i), they were observed also in bedding-parallel nodules (Fig. 10a). Host rock volumes within DBs-parallel nodules are still characterized by blocky sparite cement with some minor dark-CL growth sub zones (Fig. 10i, m, n), similarly to what was observed in bedding-parallel nodules.

350        To evaluate any sign of dissolution in nodules, we carefully checked cement crystals morphologies adjacent to poorly- or non-cemented host rock sectors at the edges of nodules. Here, cement crystals boundaries are regular and sharp (Fig. 10m, n).

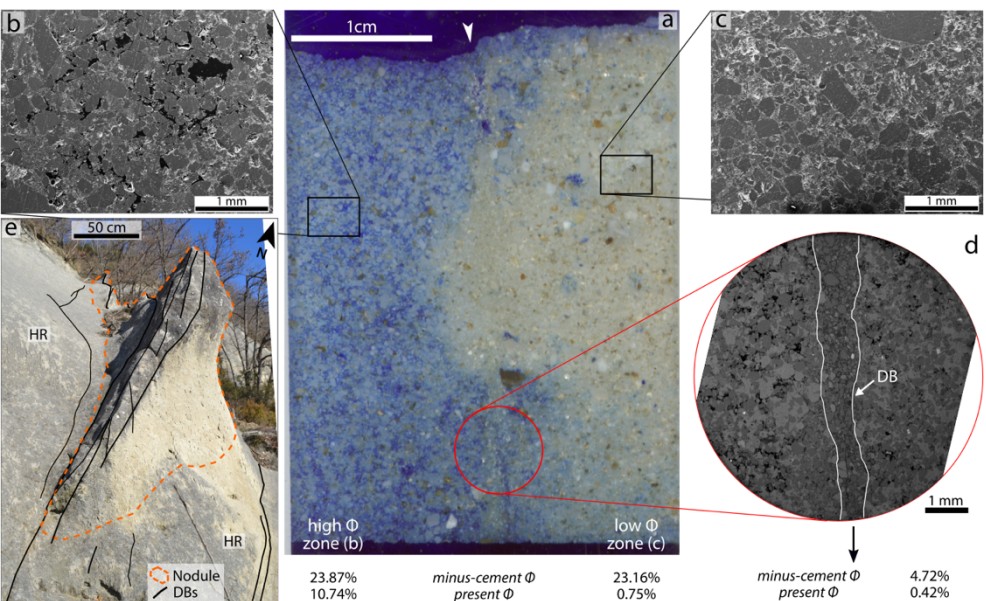

**Figure 9.** Host rock (HR) and DBs porosity and relationships between cement and DBs in Loiano Sandstones. (a) Polished thin sections impregnated with blue-dyed resin. The section shows a DB (arrow) separating two host rock sectors: in the right-hand side there is extensive calcite cementation whereas on the left-hand side cementation is poor. (b, c) Secondary electrons and (d) backscattered electrons SEM images from different sectors of the section (a). Porosity ($\phi$) estimations data in (a-d) from Del Sole and Antonellini (2019). (e) Field example where a nodule is asymmetrically placed on a side of the DBs, similarly to what is observed in (a). See text for further details.










**Figure 10.** Natural- and CL-light photomicrographs showing the microstructure, and cement textures in (a-h) bedding-parallel and (i-n) DBs-parallel nodules. (a-b, d) Microfractures forms at grain contacts due to stress concentration at contact points. (i) Microfractures are common in the host rock areas close to the DB. (b) Feldspars break mainly by cleavage-controlled intragranular fractures (white dotted line). Some framework grains show planar to slightly undulated framework grain-to-grain contacts. Detrital carbonate clasts and (c-d) bioclasts are partially dissolved at grain contacts. (e) Some detrital grains are corroded and coated or partially replaced by cement. (f) Syntaxial overgrowth cement on a detrital carbonate clast (D). Bright-CL (c) circumgranular pore-lining (dogtooth texture) and (d) intraskeletal (drusy mosaic) calcite cement was observed only in bedding-parallel nodules. The main cement phase is characterized by bright-orange to orange-CL calcite cement that fills intergranular porosity and intragranular fractures both in (a, b, e-h) bedding-parallel nodules and (i-n) DB-parallel ones. (g-h, m-n) Host rock volumes in nodules are characterized by blocky sparite to poikilotopic calcite cement with minor dark-CL sub-zones, whereas (j-l) the cement in DBs is fine-grained sparite. (m-n) At nodules edges, cement crystal rims are regular and sharp suggesting absent or negligible dissolution. Bright grains in (k, l) are detrital calcite (D). Qz - quartz; Fs - feldspar; M - mica; Rf - rock fragment; B - bioclast; φ - pore space; C - calcite cement; TL - transmitted light. See text for further details.

## 5.2 Bollène

The host sands are weakly cemented, with the exception of localized carbonate cementation described above. Host rock grains are mostly rounded and without a fabric (Fig. 11a, f). A few intragranular fractures break framework grains at the contact-point among two or more grains (Fig. 11a, b), and they are more frequent approaching the DB (Fig. 11g, h, j). These microfractures, in fact, are a common feature in the host rock sectors in proximity to the DB and they are preferentially oriented with respect (ranging from ca. 30° to 50°) to the force chain around DB (Fig. 8c). Despite the grains being fractured in the host rock sectors close to the DB, and the reduction in grain size is limited. Here, we describe the microstructure of NW-SE/NNW-SSE normal-sense and strike-slip bands, and NE-SW/ENE-WSE strike-slip bands sets. The general pattern between these two sets of DBs is similar and here we depict the main features and differences. The most recognizable features that characterize the microstructure of both DBs sets are the reduction of grain size, porosity, and a tighter packing relative to the host rock (Fig. 11). NE-SW strike-slip bands (Fig. 11i-l) have a higher degree of grain comminution, porosity reduction, and tighter packing when compared to NW-SE bands (Figs. 8c and 11g-h). Most grains within the bands are fractured and angular. Despite the strong comminution, a few rounded large survivor quartz grains are preserved in the DBs matrix (Fig. 11g, h, k). Fine angular grains that are mostly comminuted feldspars fragments and secondarily quartz and minor oxides make up the matrix. Feldspar sheared grains commonly organize in highly comminuted lenses. These stripes are well recognizable both in natural-light where they appear as brown lenses (Fig. 11g, h) and in CL where they show a bright light blue color (Fig. 11h). The preferential alignment of elongated grains and stripes of detrital feldspar produce a foliated pattern that is sub-parallel to the DB. We observed also fine particles of crushed calcite cement among the matrix grains within NE-SW bands (Fig. 11k, l). In some cases, the grains in the host rock areas in proximity to the DB are encased by relatively undeformed carbonate cement (Fig. 11i, j). Some grains in the host rock are corroded and partially replaced or coated by calcite cement (Fig. 11c).



The main cement in spherical and tabular nodules is characterized by a texture of poikilotopic spar cement that infills intergranular pores (Fig. 11a-e). While this cement is almost everywhere uniform in terms of texture, it shows some inhomogeneities in terms of CL-pattern. Most of the cement is non-luminescent (dark luminescence) under CL (Fig. 11c, d) but a few crystals show partial overgrowths with bright-orange CL color (Fig. 11e). When the crystal has a heterogeneous

CL-pattern, the non-luminescing zones are mainly in the crystal core whereas the luminescing sub-zones are mostly at the crystal edges (Fig. 11e). A very thin film (up to c. 10μ thick or less) of bright-orange CL calcite cement commonly coats the detrital grains (Fig. 11c), and it is visible also under natural-light (Fig. 11b). In the nodules, some of the intragranular microfractures at contact-point are filled by cement (Fig. 11a); a few are not (Figs. 11b). The cements described above (pore-filling and grain-coating) are relatively undeformed (i.e. no microfractures, no twin-lamellae) and still preserving the original

shape, except where the NE-SE/ENE-WSW strike-slip bands crosscut the cement nodules. At the crosscutting site, indeed, and more specifically in the host rock sectors in proximity to the NE-SW/ENE-WSW bands, we observe intragranular fractures at contact point and the onset of cement comminution between quartz clasts (Fig. 11i, j). Fine particles of crushed detrital calcite cement are found among the cataclastic matrix grains within NE-SW/ENE-WSW strike-slip bands where they interact with nodules (Fig. 11k, l). At the microscale, no preferential or significant calcite cementation was observed in

association with the NW-SE bands. The association between cements and these latter bands was observed only at the mesoscale (see Sect. 4.2).







**Figure 11.** Natural- and CL-light photomicrographs showing the internal texture and microstructure of (a-e) spherical and tabular nodules
and (f-l) DBs. (a, b, f) Host rock grains are mostly rounded and nearly undisturbed. Some microfractures break framework grains at the
contact-points. In nodules, (a) some of the microfractures are filled; (b) a few are not. (g, h) Microfractures are more frequent approaching
the DB and (j) they are preferentially oriented with respect to the band (white arrows). Cement fills intergranular pore-space and
intragranular fractures. (c-e) The major diagenetic component is poikilotopic spar cement with dominant dark-luminescence and (e) minor
bright-orange CL growth sub-zones. (c) A very thin film of bright-orange CL calcite cement often coats detrital grains. Minor diagenetic
alterations are corroded detrital grains that are partially replaced by calcite cement; see inset in (c). (g, h) NW-SE bands and (i, l) NE-SW
strike-slip bands show a similar pattern, but NE-SW bands feature a high degree of grain comminution and porosity reduction. (i, j) NE-
SW strike-slip bands crosscut cement nodules; intragranular fractures (white arrows) and incipient stage of cement comminution (black
arrows) between quartz clasts in the host rock sectors in proximity these bands. (k, l) Fine particles of crushed detrital calcite cement (D)
are found among the matrix grains at the crosscutting site. The inset in (a) is a backscattered electrons SEM image. Qz - quartz; Fs -
feldspar; φ - pore space; C - calcite cement; TL - transmitted light.

## 6 Cement stable isotopes geochemistry

### 6.1 Loiano

Results from the stable isotope analysis of the Loiano samples are presented in Fig. 12a, which shows that cement
from the nodules has $\delta^{13}C$ values between -7.68 and -1.47 ‰ (V-PDB) and $\delta^{18}O$ values between -4.42 and -1.35 ‰ (V-
PDB). Figure 12a shows two groups of data in the $\delta^{18}O$–$\delta^{13}C$ space. The group referring to the DBs-related nodules is
characterized by isotope compositions between -5.41 and -1.47 ‰ (V-PDB) for $\delta^{13}C$, and between -4.42 and -1.40 ‰ for
$\delta^{18}O$ (V-PDB). The group referring to bedding-parallel nodules has isotope compositions between -7.68 and -5.94 ‰ (V-
PDB) for $\delta^{13}C$, and between -2.09 and -1.35 ‰ (V-PDB) for $\delta^{18}O$. Both cement groups (DBs-parallel and bedding-parallel
nodules) have a relatively narrow range of oxygen isotopic composition featuring a near-vertical alignment in the $\delta^{18}O$–$\delta^{13}C$
cross-plot. DBs-parallel nodules show a slightly wider span of $\delta^{18}O$ composition when compared to bedding-parallel
nodules. However, carbon isotopic composition shows a wide range of variability, both when considering the total isotopic
composition data and when considering the cement groups data.

### 6.2 Bollène

Figure 12b reports the stable isotope analysis of the Bollène samples in $\delta^{18}O$–$\delta^{13}C$ space for two groups of data. The
cement group referring to the DBs-related nodules has $\delta^{13}C$ values between -7.73 m and -4.68 ‰ (V-PDB) and $\delta^{18}O$ values
between -7.70 and -5.88 ‰ (V-PDB). The other group is from cement sampled in a calcrete level observed within the same
Turonian sandstone few meters above the studied outcrop (see Sect. 4.2 and the Supplement for details), and it is
characterized by isotope compositions between -2.54 and -2.39 ‰ (V-PDB) for $\delta^{13}C$, and between -6.58 and -6.32 ‰ (V-
PDB) for $\delta^{18}O$. Both cement groups have a relatively similar $\delta^{18}O$ signature and a relatively narrow range of $\delta^{18}O$





composition varying only between -7.70 and -5.88 ‰ (V-PDB). In a similar way, cement sampled from the calcrete has a narrow range of $\delta^{13}C$ composition and show the heavier $\delta^{13}C$ values in the data set. However, $\delta^{13}C$ composition of DB-related nodules has a wider variability range and is the most depleted in the data set.

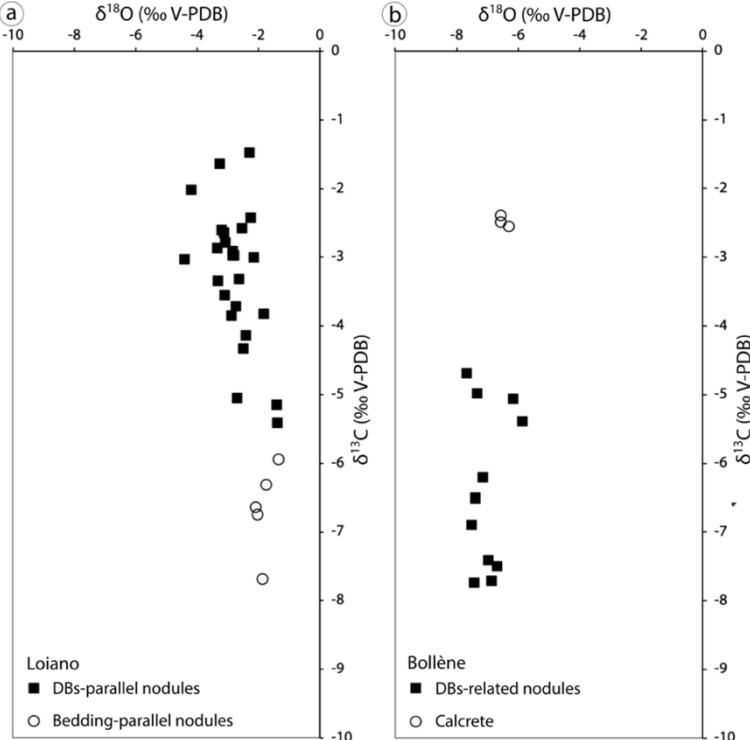

**Figure 12.** Stable isotopes analysis results. (a) Cumulative isotopic data characterizing the DBs-parallel nodules (black full-dots) and
bedding-parallel ones (empty-dots) inside the Loiano Sandstones. (b) Isotopic data from the DBs-related nodules (black full-dots) and cement sampled in the calcrete (empty-dots) in Bollène quarry (see the Supplement for details).

## 7 Discussion

In the following, we compare the two field sites highlighting their similarities and differences concerning the interaction between deformation, fluid flow, and diagenesis. We discuss the influence of DBs on fluid flow, their role in
enhancing diagenesis and localizing diagenetic products (nodules). Finally, we propose an explanation for the geochemical environment within which fluids were sourced and precipitated the nodule cement. We then explore the implications of structural diagenetic heterogeneities (SHD) upon subsurface fluid flow and reservoir characterization.

### 7.1 Deformation band characteristics



In both study areas, the key, common structural element are the DBs. In *Loiano*, geometric compatibility between

DBs strike (Fig. 2d) and large-scale normal faults in this sector of the Northern Apennines (Fig. 2a, c; Antonellini and

Mollema, 2002; Picotti et al., 2009) indicates that they might be coeval. For instance, our study outcrops are close to a high-

angle normal fault striking nearly N-S and dipping W (Del Sole et al., 2020). However, the presence of multiple sets of DBs,

with different trends (Fig. 2d), kinematics, and ambiguous crosscutting relationships (Fig. 3) suggests that the Loiano

Sandstones recorded multiple tectonic phases (Cibin et al., 2001; Antonellini and Mollema, 2002) from the Eocene onwards.

A clear sequence of deformation events has not yet been defined. In *Loiano*, the inhomogeneous mesoscale pattern and

geometry of DBs along the exposed sandstone sequence is related to variations in host rock properties, such as sorting,

porosity, and grain-size (Fig. 5e). Low porosity and/or poorly sorted coarse-grained host sandstones promote localized

deformation commonly with a single straight-trace DB accommodating all displacement. High porosity and/or well-sorted

fine-grained sandstones feature, instead, a more distributed deformation with several anastomosing DB segments, where

each segment accommodates part of the total displacement. Others host rock properties steering the overall DB

characteristics are mineralogy, lithification, and grain roundness (Antonellini and Mollema, 2002; Fossen et al., 2017; Del

Sole and Antonellini, 2019).

In the *Bollène quarry*, structural features include diachronous and differently oriented DBs and occur only in the

unconsolidated Turonian sands (Fig. 2c). Crosscutting relationships between different sets point out that ESE-WNW reverse

DBs set should be the oldest. Their kinematics, low angle planar attitude and organization in conjugate and densely

distributed networks, are typical features of DBs formed in a contractional regime. This evidence and the compatibility

between the strike of reverse DBs and larger scale E-W trending thrusts and folds in the area (Fig. 2a, b) would indicate that

they formed contemporaneously and within a single stress field (Ballas et al., 2013, 2014; Soliva et al., 2013). In particular,

these DBs are likely associated with the Paleocene to early Oligocene Pyrenean shortening (Sanchis and Séranne, 2000;

Lacombe and Jolivet, 2005) (Fig. 2a). The NE-SW/ENE-WSW strike-slip bands displace the ENE-WSW reverse DBs, the

NW-SE bands, and the NW-SE-trending carbonate cementation; hence, NE-SW/ENE-WSW strike-slip bands should be the

younger set among the observed ones. They are most probably related to the (left-lateral) strike-slip reactivation of some

major preexisting NE-SW faults in the region (e.g. Cevennes and Nîmes faults; Fig. 2a) during the Pyrenean event or in the

Miocene to Quaternary age NNE-SSW Alpine contraction (Champion et al., 2000). The existence of a conjugate set of

strike-slip bands with left-lateral (mostly ENE-WSW) and right-lateral (mostly NE-SW) kinematic (Fig. 6b) it places the

maximum-compressive stress axis ($\sigma_1$) at ca. N50°E, thus, these DBs are consistent with the Miocene Alpine contraction

(Fig. 2a). Two hypotheses can explain the occurrence of NW-SE/NNW-SSE DBs associated with cement nodules. They

could have been formed as dextral strike-slip structures (Figs. 6b and 7f) during the NNE-SSW Pyrenean-Provencal

shortening (Fig. 2a). Some of these faults could have been reactivated as normal faults during the NW-SE Oligocene-Early

Miocene extension. A second possibility is related to the presence of a NE-SW map-scale normal fault, the Bollène Fault

(Saillet, 2009), close to the outcrops discussed in this study (Fig. 2c). In this framework, the NW-SE set of normal DBs



could be genetically linked to a stress perturbation (in a dilation jog or quadrant) around the large-scale Bollène fault, during its Oligocene activity.

Microstructural observations indicate that DBs in both *Loiano* and *Bollène,* developed by mechanical grain
fracturing and compaction with minor contributions by shear-related grain disaggregation by rolling and grain boundary sliding. Zone of bands show a similar pattern but the degree of cataclasis is higher than single DBs. These structures can be classified as compactive shear bands (CSB) with cataclasis (Aydin et al., 2006; Fossen et al., 2017).

## 7.2 Cement distribution and its relationship with deformation bands

The distinctive feature of the *Loiano Sandstones* is a spatially heterogeneous cementation in the form of nodules.
Field evidences indicates that DB formation predates calcite cementation. All nodules are spatially related to DBs (Figs. 3 and 5a-c) except for those that are situated along bedding planes (~25% of the total nodules; Figs. 3 and 5d, e). In contrast, not all DBs are associated with nodules. A clear correspondence always exists between shape and elongation direction of the nodules and the DBs direction. This pattern is observed also from aerial photographs (Del Sole and Antonellini, 2019). Localized of cementation along these structural features itself is an indication that deformation preceded cement precipitation
(e.g. Eichhubl et al., 2004), unless the sandstones were completely cemented first, and then completely removed except from the DBs. In this case, the cement in the DBs cataclastic matrix would have been likely preserved with orange colors; but nothing like that was observed. Moreover, cement morphologies adjacent to the porosity at nodules edges suggest that cement dissolution has not occurred (Fig. 10m, n), thus excluding that nodules, both those parallel to bedding and those parallel to DBs, are relicts from an overall dissolution process. No DBs crosscut the cement, at least for those sets that are
spatially related to nodules (NNW-SSE to NE-SW) suggesting that cementation postdate DBs development. The precipitation of cement and (the consequent) lower porosity would favor the formation of joints over DBs in the sandstone (Flodin et al., 2003; Aydin et al., 2006; Fig. 5a, b). The presence of pore-filling cement would increase the strength of the sandstone (Del Sole et al., 2020), preventing rotation and sliding of particles, increase rock cohesion (Bernabé et al., 1992) and grain contact area, thus yielding a uniform contact stress distribution and higher stiffness (Dvorkin et al., 1991).
Extensive cement, then, would inhibit DBs development. We can affirm, hence, that cement overprints the deformation bands.

Results from microstructural observations show that intergranular cement in the nodules encloses the grains both within host rock and DBs, it overprints burial-related mechanical and chemical compaction features (Fig. 10a-h). These evidences point out that the formation of authigenic cements occurred after significant compaction (Cibin et al., 1993;
Milliken et al., 1998). Estimated burial depths (in meters) referred to the top of the Loiano Sandstones are 800-1000 (Cibin et al., 1993) and 700-1200 (McBride et al., 1995). Transgranular microfractures at grain contacts are due to stress concentration at contact points and they are interpreted as load-bearing structures within the granular framework (e.g. Antonellini et al., 1994; Eichhubl et al., 2010; Soliva et al., 2013). In DBs-parallel nodules samples the cement that fills the



transgranular fractures is in continuity (i.e. same textural and CL characteristics) with the pore-filling cement outside the

fractures. The presence of undeformed cement within structural-related features such as microfractures and crushed grains (Fig. 10i-l), both within and outside the DBs, proves that cement precipitation occurred after (at least after the early stages of) deformation.

While the bands are the main controlling factor in DBs-parallel nodules (location, geometry and elongation direction), the occurrence and location of bedding-parallel nodules is controlled by grain-size and contrast in grain-size

within the host rock. Although bedding-parallel nodules are found in all sands, they are more common within coarse-grained levels (≥ medium sands; i.e. size range: 0.25-0.5 mm). There are no nodules in sediments below the sand range or in layers with permeability below 100 mD (Del Sole et al., 2020). Moreover, bedding-parallel nodules are often restricted to sand level in contact above and/or below with clay/silty levels (Fig. 5d, e). Hence, grain size, and permeability variations are the most important factors controlling diagenetic and nodules formation in Loiano. The grain size and permeability variations as

dominant control on nodules development in porous media is also reported by other authors (Mozley and Davis, 1996; Davis et al., 2006; Cavazza et al., 2009; Balsamo et al., 2012). In general, bedding-parallel nodules show a more rounded morphology when compared to DBs-related nodules. The former nodule type owns its smooth morphology to a homogeneous and isotropic weathering; the sharp and squared shapes of DB-s parallel nodules is probably due to the anisotropy introduced by the DBs in the host rock that influences the cementation. The interplay between band strength and

erosion may also be influent on nodule shape.

In the *Bollène quarry*, all calcite nodules occur in association with the DBs, and in particular with the NW-SE/NNW-SSE set (Fig. 6b). At this site, we observe complex relationships among multiple deformational events and diagenetic ones. Timing of bands and nodules is inferred from crosscutting relationships. There are no evidences of low-angle ESE-WNW reverse-sense DBs crosscutting the cement nodules, whereas NE-SW to ENE-WSW trending strike-slip

DBs offset the reverse-sense bands, the NW-SE bands, and the NW-SE-trending cement nodules (Figs. 6b and 7). Regarding on the relative timing between NW-SE bands and the nodules, the same argument made for Loiano (see above in this Sect.) holds for *Bollène*, with the exception that NW-SE-trending nodules and DBs are not superposed. Deformation bands are always overprinted by cement but the spatial overlap between DBs and nodules (Fig. 7e) is unusual. In *Bollène*, nodules occur in compartments that are spatially confined by DBs zones. Tabular nodules and clusters of spherical nodules are

oriented with the major axis parallel to the NW-SE DBs (Figs. 6b and 7a). The parallelism and localization of cementation along these structural features could be, by its own, an indication that deformation predates cement precipitation. The NW-SE bands do not crosscut the cement. Cement precipitation, therefore, is a post-kinematic process. In the proposed relative timing of events, calcite cementation occurred between the NW-SE bands formation (Pyrenean contraction or Oligocene-Miocene extension?) and the NE-SW strike-slip bands (Miocene-Quaternary age Alpine shortening?). Microstructural

observations show that the dominant phase of calcite cement (intergranular) encloses the grains within the nodules and it overprints only a proportion of the transgranular microfractures at grains contact points. As far as we can say, all microfractures in the nodules are filled by a cement that is in continuity (i.e. same texture and CL characteristics) with the



cement outside the grain (i.e. pore-filling cement). Unfilled microfractures (Fig. 11b), most probably, were not connected to the poral network and they were potentially quickly isolated by the calcite mineral growing in the pore space. It is less likely that they formed after cement precipitation, otherwise the cement would have been broken.

**7.3 Role of deformation bands on fluid flow and diagenesis**

The localized diagenesis observed in form of nodules at Loiano and Bollène provides evidence for the effect of structural heterogeneities, such as DBs, on fluid flow in porous sandstones (Eichhubl et al., 2004, 2009; Balsamo et al., 2012; Philit et al., 2015; Del Sole and Antonellini, 2019; Pizzati et al., 2019; Del Sole et al., 2020). The petrophysical properties (e.g. porosity, permeability, capillary entry pressure) of DBs influence fluid flow and localize diagenesis (e.g. cement precipitation).

A marked grain-size reduction, grain-surface roughening, porosity, and pore-size reduction characterize the DBs presented in this work. In *Loiano*, the combined effect of cataclasis and compaction in the DBs causes porosity reduction by one order of magnitude, permeability reduction by three orders of magnitude, and advective velocity reduction by 2 orders of magnitude with respect to the host rock (Del Sole and Antonellini, 2019). Similarly, DBs in the *Bollène* quarry have lower permeability (up to 3 orders of magnitude) and porosity (up to 50%) (Ballas et al., 2014) when compared to the host rock. Several authors recognized that cataclastic DBs negatively affect flow tortuosity in reservoirs and produce capillary barriers that severely baffle the flow at the reservoir scale and limit cross flow between host rock compartments (Harper and Mofta, 1985; Edwards et al., 1993; Lewis and Couples, 1993; Antonellini and Aydin, 1994; Leveille et al., 1997; Gibson, 1998; Antonellini et al., 1999, 2014; Sternlof et al., 2004; Rotevatn and Fossen, 2011; Ballas et al., 2012; Medici et al., 2019; Romano et al., 2020). Smaller pores within bands result in higher capillary forces than in the host rock. This may cause higher water saturation within the bands with respect to the host rock (Tueckmantel et al., 2012; Liu and Sun, 2020). Higher degree of flow tortuosity (reduction in pore interconnectivity) and lower porosity and permeability within the bands may increase the fluid retention time regardless of the water-saturation conditions (Antonellini et al., 1999; Sigda and Wilson, 2003; Wilson et al., 2006). Recently, Romano et al. (2020) documented with single and multiphase core flooding experiments, that cataclastic bands can strongly influence the fluid velocity field. Other authors (Taylor and Pollard, 2000; Eichhubl et al., 2004) recognized that a slower rate of solute transport relative to the fluid within the bands causes the formation and local perturbation of diagenetic alteration fronts. In light of these considerations and the temporal and spatial relationships between bands and cements obtained from field and microstructural observations, we propose a model for selective cement precipitation associated with DBs. In our model we assume a reservoir in saturated condition (see also Sect. 7.4). We think that the presence of the bands in *Loiano* slowed down the fluid flow. This happens because the bands have lower permeability and transmissibility, a higher degree of tortuosity (i.e. lower pore size) and reduced section area available for flow (i.e. lower porosity) with respect to the host rock (Del Sole and Antonellini, 2019; Del Sole et al., 2020). Cataclasis has competing effects on advective flow velocity; it causes (i) an increase of flow velocity linked to the porosity reduction





and (ii) a decrease of the hydraulic conductivity (if the hydraulic gradient does not change). The decrease of hydraulic
conductivity (3 orders of magnitude; Del Sole and Antonellini, 2019) on reducing advective flow velocity is prevalent on the
flow velocity increase caused by porosity reduction (1 order of magnitude; Del Sole and Antonellini, 2019). As a result,
there is a net decrease in advective flow velocity in the DBs. This "slow down", alone, could represent one of the first trigger
to promote preferential cement precipitation within the band or in its proximity as observed in the field.

595         A second mechanism responsible for cement nucleation in association with DBs would be the presence of highly
reactive crushed and pervasive fractured siliciclastic grains within the cataclastic DBs (e.g. Lander et al., 2009; Williams et
al., 2015). The comminuted material of the DBs owns a large amount of reactive surface area (nucleation spots) and very
tiny pores spaces among the crushed grains. With these conditions, cement precipitation requires less free energy to occur
(Wollast, 1971; Berner, 1980) whereas greater cement abundances (e.g. Walderhaug, 2000) and faster rates of cement
emplacement (Lander et al., 2008; Williams et al., 2015) are promoted. This mechanism has already been proposed to
explain the presence of cement within the bands pore space and contrast in the degree of cementation between the bands and
the surrounding rock (e.g. Antonellini et al., 1994; Knipe et al., 1997; Fisher and Knipe, 1998; Milliken et al., 2005; Philit et
al., 2015; Del Sole and Antonellini, 2019; Pizzati et al., 2019). Indeed, this mechanism may be relevant for *Loiano* where the
calcite cement fills small pore spaces among fresh quartz and feldspar surfaces created during fracturing. This process can
explain why in most cases DBs are more cemented than the surrounding host rock. The low-porosity angular fine-grained
cataclastic matrix within the bands offers a lower energy barrier for cement nucleation (Wollast, 1971; Berner, 1980), so that
it is not necessary for fluids or brines to reach carbonate saturation for cement precipitation at grain contacts (De Yoreo and
Vekilov, 2003). On the contrary, this process was less relevant in the *Bollène quarry* where the bands are not cemented by
carbonate, and the cementation is localized in compartments between zones of bands rather than within them.

610         A third mechanism could have worked in conjunction with the presence of more reactive fine-grained comminution
products to promote cementation in the DBs. According to their experiments on analog fault gouge, Whitworth et al. (1999)
suggested a membrane behavior for faults in sandstones during cross-fault flow and solute-sieving-aided calcite
precipitation. A membrane effect and solute-sieving by faults may locally increase the concentrations of components needed
for calcite cementation (e.g. Ca and bicarbonate) and induce precipitation. In our opinion, the DBs could have acted as a
semipermeable membrane in baffling chemically reactive flow and favor cement precipitation. This process may explain a
higher concentration of cement along the DBs in nodules, and the asymmetric distribution of cement on one side of DBs
(upstream side; Figs. 4c, e and 9a, e). An analogous mechanism was proposed by other authors to explain the occurrence of
preferred and asymmetric distribution of the authigenic alterations (carbonate and clay cements, Eichhubl, 2001; hematite
bleaching, Eichhubl et al., 2004) on the upstream side of DBs in sandstones. Other factors that may have locally favored (the
initiation of) calcite cement precipitation, are the growth of cement on detrital grains (*Loiano*, Fig. 10e; *Bollène*, Fig. 11c)
and the presence of broken detrital carbonate clasts (e.g. shell fragments) that act as a "seed" (cement nucleation sites) (e.g.
Bjørkum and Walderhaug, 1990). The latter case was observed in *Loiano*, mainly in bedding-parallel nodules (Fig. 9c, d).
The proposed mechanisms explain how and why cement precipitation would occur within the band and in its proximity, as



observed in the field. Our hypotheses confirm the theoretical and flow simulations as well as the analog experiments which
demonstrated that DBs can negatively affect the fluid flow in porous sandstones (e.g. Rotevatn and Fossen, 2011;
Antonellini et al., 2014; Romano et al., 2020) and enhance cement precipitation (Whitworth et al., 1999).

**7.4 Structural diagenesis scenario for carbonate nodules formation**

Here, we will integrate the petrographic observations and stable isotopes characterization of cements with the meso-
scale spatial organization and the micro-scale textural relationships between nodules and DBs to discuss the geochemical
conditions and potential fluid sources that controlled the formation of carbonate nodules in the studied areas.
In *Loiano*, the first calcite cement would be the intraskeletal and pore-lining cement associated with bioclasts in bedding-
parallel nodules (Fig. 9c-d). The cement fabric and textures, circumgranular dogtooth and void-filling drusy mosaic, and the
dominant bright-orange CL of calcite associated with bioclasts could suggests a meteoric phreatic environment with
reducing conditions (low $pO_2$) (Longman, 1980; Moore, 1989; Hiatt and Pufhal, 2014; Adams and Diamond, 2017). Drusy
calcite spars can result from replacement of aragonite in bioclasts that occur in meteoric environments (e.g. Flügel, 2013).
The second, more pervasive, phase is the intragranular cement observed in all the nodules. The texture and fabric, a mosaic
of blocky sparite with coarse crystals and homogeneous distribution, still point to meteoric phreatic conditions (Longman,
1980; Flügel, 2013; Adams and Diamond, 2017). The invariably negative, depleted $\delta^{13}C$ and $\delta^{18}O$ values found in both types
of nodules are consistent with a meteoric environment of precipitation (Nelson and Smith, 1996). The dominant bright
cathodoluminescence response suggests an environment with reducing (low $pO_2$) geochemical conditions. The zonation
could be induced by changes in the water geochemistry caused by Eh and pH oscillations (Hiatt and Pufhal, 2014). It could
indicate the shifts from more reducing (bright CL) to more oxidizing (dark-CL) conditions and vice versa (Barnaby and
Rimstidt, 1989) due to water table fluctuations (Li et al., 2017). The intergranular cement pattern described above is
analogous in DBs-parallel nodules and bedding-parallel nodules meaning they probably formed in a similar meteoric
phreatic environment. Phreatic meteoric conditions for the nodules formation point to a shallow diagenesis, and it is
consistent with the shallow burial depths estimated for the Loiano Sandstones (see Sect. 7.2). DBs-parallel and bedding-
parallel nodules show a similar composition for $\delta^{18}O$, however bedding-parallel nodules have more depleted $\delta^{13}C$ values.
This could reflect a higher contribution of organic carbon from soil-derived $CO_2$ (Hudson, 1977), and could indicate that the
bedding-parallel nodules formed in shallower conditions with respect to DBs-parallel nodules. Another explanation could be
that the two types of nodules were formed by different episodes of water income with different (external) environmental
conditions. The difference in isotopic composition between these two types of nodules, indeed, suggests different cement
precipitation timing and water compositions as proposed by McBride et al. (1995) and Milliken et al. (1998). Bedding-
parallel nodules organization within the local rock sequence supports a phreatic condition for these nodules. Clay levels are
considered sealing features at least at short-time scales, so that the formation of bedding-parallel nodules in sand levels
below and/or confined in between clay levels let us infer that fluid circulation was not occurring in the vadose zone.



Bedding-parallel nodules, then, could have formed by lateral fluid circulation in saturated conditions (Fig. 13a). The occurrence of preferential cementation in coarser sediments could be another macroscopic evidence for phreatic cementation, and it is due to higher saturated permeability and larger solute flux (Mozley and Davis, 1996). DBs-parallel nodules, however, indicate that local fluid circulation was mostly driven by these structural heterogeneities. The asymmetric distribution of cement in some nodules associated with DBs could be also explained by lateral fluid circulation (Fig. 13a), and the accumulation of cement would result on the upstream side of the DBs (Fig. 4c, 9, and 13a). In other cases, cement is roughly symmetrical with respect to the bands, or it is placed in the zones where conjugate bands intersect (Figs. 4b, 5c, and 13a). Cementation pattern could be used to infer the paleo-fluid flow direction at the time of calcite precipitation (Mozley and Goodwin, 1995; Mozley and Davis, 1996; Cavazza et al., 2009; Eichhubl et al., 2009; Balsamo et al., 2012). The different spatial arrangements between DBs and nodules in Loiano make the paleo-fluid flow direction reconstruction complex. In our opinion, the most likely interpretation is that both lateral flow in saturated conditions and "direct" meteoric infiltration from the surface, and percolation through the rock succession, contributed to the formation of nodules in Loiano (Fig. 13a). In this view, the calcite source (i.e. diffusive supply of $Ca^{2+}$ and $HCO^{3-}$) could derive from the infiltration of $CaCO_3$-saturated meteoric fluids carrying soil-derived $CO_2$ (Hudson, 1977; Nelson and Smith, 1996), and/or it could be autochthonous deriving from detrital carbonate grains in the sandstone layers or intra-formational deriving from shale beds and calcite-rich clays layers (McBride et al., 1995; Milliken et al., 1998). In both the scenarios the water precipitates calcite where fluid flow slows down in proximity to zones of DBs (DBs-parallel nodules) and low conductivity layers (bedding-parallel nodules). McBride et al. (1995) suggest that the precipitation of calcite along faults (DBs) in *Loiano* was induced by the mixing of locally derived formation water with meteoric water introduced along the faults or by a loss (exsolution) of $CO_2$ along the fault zones. These mechanisms, however, assume that DBs were conductive to fluid flow. This hypothesis disagrees with our measurements of the DBs hydraulic behavior. More likely, carbonate DBs cementation could have resulted by $CO_2$-saturated groundwater (Fig. 13a). We cannot exclude the role of active normal faults in the area (Picotti and Pazzaglia, 2008; Picotti et al., 2009; Fig. 2a), that might have had a control on regional subsurface fluid circulation. These faults could have cut through top/bottom seals and driven fluid migration from top/underneath aquifers (Fig. 13a). Episodic fault activity could also favor (episodic) horizontal fluid migration along layering at the time of faulting, and it could explain the occurrence of nodules (Fig. 13a) and their different isotopic signature (Fig. 12a). From our observations, therefore, we can say that the selective cementation process in the Loiano Sandstones depends on "regional" hydrological factors (e.g. topographic gradient, bedding, faults?) coupled, locally, to the presence of DBs.

In the *Bollène quarry*, the relative timing of DBs formation and cementation in the Turonian Sandstones is complex to unravel. Carbonate cementation, in fact, occurs in between distinct deformation phases with multiple DBs formation. The dominant dark cathodoluminescence pattern and homogeneous distributed poikilotopic spar texture suggests an oxidizing (high $pO_2$) meteoric phreatic environment (Longman, 1980; Moore, 1989; Flügel, 2013; Hiatt and Pufhal, 2014). The zonation is induced by unsteady geochemical conditions, i.e. changes in water geochemistry, and the bright-orange CL growth sub-zones could be related to stages with more reducing conditions (Barnaby and Rimstidt, 1989; Hiatt and Pufhal,



2014). The invariably negative, depleted $\delta^{13}C$ and $\delta^{18}O$ values of nodules in Bollène are consistent with a meteoric environment in a continental setting (Nelson and Smith, 1996). Maximum burial depth of the Turonian sandstone was estimated, through stratigraphic constrains, to be $400 \pm 100$ meters (Ballas et al., 2013; Soliva et al., 2013). This data supports the shallow conditions for nodules diagenesis in Bollène. The phreatic environment is more probable given that in vadose conditions we should have observed meniscus cements, and because massive calcrete such as observed in the study

area generally form in groundwater environment (e.g. Alonso-Zarza, 2003). In the vadose zone, besides, DBs would enhance unsaturated flow relative to the host rock (Sigda et al., 1999; Wilson et al., 2006; Cavailhes et al., 2009; Balsamo et al., 2012). Field evidences suggest that clusters of low permeability DBs in *Bollène* impeded cross-fault flow since no cement was found in superposition with the DBs. The presence of nodules in between the DBs clusters most likely implicates that the DBs forced the fluid flow and localized the diagenesis in parallel-to-bands compartments. These evidences and the fact

that nodules are homogenous along their elongation direction discredit the hypothesis of lateral flow. The cement could have been originated from a (i) downward fluid flow directly from infiltration of meteoric waters, or (ii) an upward flow of basinal fluid (pressurized aquifer) along fractures and faults pathways (Fig. 13b). In both cases, the water flow was potentially driven from the vertical continuity of DBs clusters, that have acted as the propagation features of faults in overlying (i) or underlying (ii) series and aquifers (Fig. 13b). This scenario could explain why the cement is found only in

association with the NW-SE DBs. In both cases (i) and (ii), the source that could provide the constituent necessary for the precipitation of cement in nodules (i.e. Ca and bicarbonate) would come from the surrounding carbonates. Above the Turonian sandstones there are several carbonate layers in the Upper Turonian and Santonian interval (Fig. 2c; Ferry, 1997), whereas below there are carbonates belonging to the Jurassic and Cretaceous series (Fig. 2b, c; Debrand-Passard et al., 1984). In the first case (i) continental meteoric waters saturated with meteoric carbon dioxide could have dissolved the

necessary constituents along their path through the rock succession toward the high-porosity Turonian Sandstones. The water percolation through the soil could have favored fluids acidification. Similar depleted $\delta^{18}O$ values between nodules and cement from the calcrete level (Fig. 12b) could support the (i) hypothesis, and they may have originated from similar surficial cement source from downward water flow in association with variations of bicarbonate concentration and/or pH in the water table. In the second (ii) hypothesis nodule cement resulted from $CO_2$ exsolution during the upward flow of basinal

brine or $CO_2$-saturated groundwater in a pressurized aquifer. In both cases, the flow path is probably controlled by the fractures and faults in the carbonate rocks (Fig. 13b).







**Figure 13.** Generalized conceptual model for calcite nodules precipitation in the two study areas: (a) Loiano and (b) Bollène. See text
(Sect. 7.4) for details. The inset sketches in (a) show possible paleo-fluid flow direction at the time of calcite precipitation for different
"DBs-nodule" configurations.





**7.5 Implications for subsurface fluid flow, reservoir characterization, and resources development**

Models for calcite cementation are of fundamental importance for predicting sandstone and fault-rock properties such as porosity, permeability, compressibility, and seismic attributes. In Loiano, for example, zone of DBs has acted as
fluid flow baffles steadily. First, they slowed the fluid flow and localized cement precipitation, acting as areas of preferential cementation in otherwise excellent porosity sandstones. The resulting diagenetic products increase porosity and permeability reduction caused by cataclasis, further affecting subsequent fluid circulation. Now, the presence of structural-related cement in the form of concretions i) strengthens the rock volume, ii) degrade porosity and permeability increasing the sealing capacity of zone of deformation bands, and iii) impart mechanical and petrophysical anisotropy to the host rock (Del Sole et
al., 2020). Therefore, we think that it is important to consider the possibility of concretions to form in association with faults within siliciclastic reservoirs, especially where these structures (DBs) are below seismic resolution (e.g. Del Sole et al., 2020). It is also critical to understand structural diagenesis heterogeneities (SHD) spatial organization, extension, continuity, density, their hydraulic role in terms of fluid flow circulation as well as their mechanical influence on the host rock. This information should be included in a robust fault seal analysis and, in general, it is beneficial during geofluids exploration and
energy appraisal, resources development strategies (groundwater, geothermal, hydrocarbon), well production, reservoir simulation modeling, geomechanical evaluation of a drilling site, and other environmental and industrial operations (e.g. waste fluid disposal; groundwater contaminants; geologic $CO_2$ sequestration; Enhanced Oil Recovery [EOR]). When the cementation is heterogeneous, such as in the examples presented in our work, it could be difficult to model and predict, especially when data are spatially discontinuous (e.g. wells). In these cases, outcrop-based studies allow continuous and
more reliable reconstruction of the cement distribution. The characterization of SDH network distribution (e.g. Del Sole et al., 2020) allows to predict where (i.e. location and volumes) and how (i.e. spatial organization) the reservoir compartments are arranged and how the fluid circulation can be affected.

**8 Conclusions**

In this contribution, we show two examples of structural control exerted by deformation bands on fluid flow and
diagenesis recorded by calcite nodules strictly associated with deformation bands. The objective of this research was to constrain the impact of deformation bands on fluid flow pattern and their role in fostering and localizing cement precipitation in porous sandstones, as well as to elucidate the mechanisms involved in these processes. The major results of our study can be summarized as follow:

(1)    In both case studies, at least one set of DBs precedes and controls selective calcite cement precipitation in the form of nodules. The postdating localization of cementation along these structural features results in a complex and spatially heterogenous cementation pattern (SDH).



(2) Selective cementation of nodules associated with DBs indicates interaction between deformation structures, fluid flow, and chemical processes. The volumetrically significant presence of cement (10-25% of the exposed outcrops volume) indicates that fluid flow and mass transport have been strongly affected by the presence of low permeability DBs.

(3) Three main processes are proposed to explain selective carbonate cementation associated with low permeability DBs. (i) The DBs slow down the advection velocity and promote cement precipitation in the low velocity zone. (ii) Solute-sieving across the DB (membrane effect) promotes Ca and bicarbonate concentration increase on the upstream side. (iii) The high concentration of nucleation sites on the fine-grained comminution products with increased reactive surface area of the pore-grain interface in the DB triggers cement precipitation and fast growth rates. Our hypotheses are supported by field and microstructural observations and petrophysical data.

(4) In *Loiano*, mechanisms (i) through (iii) of bullet (3) likely contributed to selective cement precipitation within the bands and in their proximities, and asymmetric cement distribution with respect to the bands. In *Bollène* no clear superposition among bands and cement was observed and only the first mechanism (i) applies. Here, the clusters of bands acted as hydraulic barriers to cross-flow, thus compartmentalizing the fluid circulation and localizing diagenesis in volumes arranged parallel to the bands.

(5) In both areas, cement texture and cathodoluminescence patterns, and their invariably negative, depleted $\delta^{13}$C and $\delta^{18}$O values suggest a shallow meteoric environment for nodules formation, in phreatic conditions.

(6) In a framework of late-stage diagenesis (post-DBs formation) and saturated conditions (meteoric phreatic environment), the processes commonly employed to explain focused fluid flow and preferential cement precipitation associated with DBs, such as "transient dilation" and "capillary suction" (see Sect. 1), appear to be less pertinent and only limited to specific conditions. In Bollène and Loiano the DBs buffered and compartmentalized the fluid flow and localized the diagenesis.

(7) Further analyses, such as flow simulations and cement precipitation modeling, are deemed necessary to further explore micro-scale fluid flow and diagenetic mechanisms that drove preferential calcite cement precipitation along DBs in porous sandstones.

(8) The association of diagenetic cementation with DBs further increases the flow buffering potential of these structural features. It also creates structural diagenetic heterogeneities that impart a mechanical and petrophysical anisotropy to the host rock volume and can seriously affect the subsurface fluid circulation in porous sandstones.

*Data availability.* Most of the data produced and used to write the paper are contained in it. More detailed information will be made available on request by contacting the corresponding author.

*Author contribution.* The paper was conceived by MA and LDS. LDS collected and processed field and laboratory data, provided their interpretations, drew the figures and wrote the manuscript. All authors actively participated in the fieldwork, in discussing the results and drawing the conclusions, and critically revised the manuscript.



*Competing interest.* The authors declare that they have no conflict of interest.

*Acknowledgments.* LDS kindly acknowledge N. A. Vergara Sassarini for fruitful discussion concerning the interpretation of cathodoluminescence imaging data and M. Pizzati for technical support during sampling for stable isotope analysis. The Laboratoire Géosciences Montpellier at the University of Montpellier is acknowledged for hosting LDS as a visiting PhD student in the period between April and July 2019, during which the cathodoluminescence analysis and the fieldwork in
*Bollène quarry* were carried out. The authors wish also to thank P. Iacumin, E. M. Selmo, and A. Di Matteo for stable isotope analysis at the SCVSA Department at the University of Parma. This research is part of a PhD project of the first author. LDS dedicates this work to the loving memory of Antonio Del Sole.

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
