# Peer review of "Structural control on fluid flow and shallow diagenesis: Insights from calcite cementation along deformation bands in porous sandstones"

_Solid Earth, 2020_

## Referee Comment (RC1) · Geoff Rawling (Referee) · 4 Jul 2020

**Review of Del Sol eet al 2020 "Structural control on fluid flow and diagenesis:…"**

**By Geoffrey Rawling**

This is a well –written paper that summarizes incredibly detailed field observations of the relationship between deformation bands (DBs) and cementation. The figures are very well done. There are only a few comments on language and grammar.

My main comments on this paper is that it needs restructuring for brevity and focus or change in emphasis. As it is the paper is a good discussion of two interesting field sites and a reasonable model for how the structures and cementation at these sites formed. However, as a paper in an international journal, I think it needs to be more broad.

I think there needs to be some explicit statements in the introduction and conclusions about what new insights are provided by this study and how they are relevant to outstanding questions relating to the control of fluid flow by deformation bands. Are fundamental questions being answered? Are ideas proposed elsewhere given a more robust foundation? I think a focus on how the observations presented are relevant to, or support, proposed processes for cementation as discussed in section 7.3 would be good. But prior to all of this, there needs to be some discussion of what the outstanding questions are and why the reader should care about them.

I think these changes would give the paper much more focus and attract the readers interest immediately. The observations in the paper should be limited to those that are relevant to these questions and or are genuinely new. The maps and thin section figures are impressive and the descriptions complete and accurate but I found myself saying "What is new here? Is all this detail needed?" Many papers have contained such detailed observations of deformation bands and cements. What is essential to make the point?

As explained below some of the conclusions are vague. What data could help strengthen them? Recommendations are made that other workers should use the types of data presented here. How? A discussion of these two points would also strengthen the paper in my opinion and give it a wider audience.

As an overall recommendation I suggest accept with moderate to major revision. Perhaps resubmittal and another review is necessary if there are major changes.

Line numbers:

50. "Fluid flow mechanisms…." I don't agree. The next 20 plus lines describe numerous studies addressing the effects of dbs on fluid flow, so it seems fairly well understood to me and heavily

studied. And see the discussion and number of references listed in section 7.3. To justify the sentence in quotes I recommend stating clearly and explicitly exactly what is not currently understood and how the present study addresses and clarifies these problems.

Figures 1 and 2. Clarify what is meant by DB azimuth. Is this strike or dip direction? The azimuths range from +- 90 relative to what geographic direction?

171. I would find it much clearer if you would use the average strike or dip direction to refer to the DBs rather than EESSWWNNNWWE. Sorry, that's what it looks like on the page! You have calculated the mean orientations of the distributions in Figs 1d and 2d, so you could use them in the text.

244. Same comment as line 171. Reference to Figure 11b here should probably be 6b?

305. The microstructural observations that follow here are exhaustive. To my mind all of these features have been described elsewhere in studies of DBs. What is new here? What is relevant and essential to the main points and arguments of the paper? If it is not relevant then it can go in a supplement and or briefly summarized. It seems to me that many of the microstructural observations could be replaced by permeability data and other hydrogeologic data by these authors (from their other papers?) or summarized from the literature. Such data are much more relevant to the hydrogeological model proposed in Sections 7.3 and 7.4.

591. Do you mean to say"…decrease of hydraulic conductivity dominates over the flow velocity increase caused by porosity reduction……."

624. I don't think hypotheses is the right word here. The field observations tend to confirm the theoretical flow simulations and experiments etc. I think this should be emphasized more in the paper overall and mentioned in the introduction. The three mechanisms described in this section have been invoked in other field studies and/or examined in the laboratory, and field observations in this study suggests that they are all relevant and or viable as possible explanations for the cement distributions.

634. it would be good to add a line or two somewhere about basic interpretation of cathodoluminescence colors for those who aren't familiar.

676. Add a reference to the measurements of DB hydraulic properties, or better , add the data to the paper as suggested above.

700. Can you expand on this? Are there any microstructural characteristics of the cement that would allow interpretation of its growth direction?

685 – 715. In general this discussion about the Bollene site seems reasonable but not definitive and rather underwhelming. E.g., Statements like " most likely" and "probably". Can you rework it to say what is known about the flow patterns and diagenesis definitively versus not. What about the three mechanisms described in section 7.3. Which do you think are applicable here (and at the Italy site), or would that just be total speculation? What additional info would be necessary to better understand the flow patterns and how they are controlled by the DBs and the resulting cement diagenesis. Ie, what could be a future research direction here?

726. I would say enhance rather than increase porosity reduction etc….

733. How would you include this information in a fault seal analysis. Give an example…it seems some sort of upscaling would need to be involved. E,g., Spatial density or proportion of cemented rock per unit area or length of fault? Something else?

740. Again, give an example. If someone is working with seismic data to do a fault seal analysis or reservoir engineering study how does the present study help them predict where the reservoir compartments are arranged etc. Elaborate.

756. As noted above, I would say that the field observations support three mechanisms that have been proposed previously as relevant to the precipitation of cements around DBs.

763. Comment at line 685 in the text, apparently I missed this discussion in the text. Perhaps that section can be rewritten to more clearly state this. But then there should be some discussion as to why the difference between the applicable mechanisms at the two sites. Something about the host rocks, db microstructures, regional geology, regional flow patterns during cementation etc

---

## Referee Comment (RC2) · J. Evans (Referee) · 17 Jul 2020

More Terms ; clS

in site? geo

[referee-annotated manuscript omitted]

---

## Author Comment (AC1) · 5 Aug 2020

**Response to Geoffrey Rawling (Reviewer #1) Comments**

**Ref: se-2020-81**

**Title: Structural control on fluid flow and shallow diagenesis: Insights from calcite cementation along deformation bands in porous sandstones**

**Journal: Solid Earth**

| Major comments | |
|---|---|
| **Comment** | **Response author** |
| This is a well –written paper that summarizes incredibly detailed field observations of the relationship between deformation bands (DBs) and cementation. The figures are very well done. There are only a few comments on language and grammar. | We appreciate very much the comments of the reviewer and the help to improve our paper.

Suggested edits have been implemented and they are tracked in the revised manuscript.
Please, consider that the track-changes Word tool created some problems with line numbering and its jumping on the annotated manuscript. Line numbering of the "revised manuscript version with changes tracked" and "manuscript without tracked changes" may not coincide. The line numbering we use in this document (Author Response to Reviewer #1) refers to the revised manuscript with tracked change file. |
| My main comments on this paper is that it needs restructuring for brevity and focus or change in emphasis. As it is the paper is a good discussion of two interesting field sites and a reasonable model for how the structures and cementation at these sites formed. However, as a paper in an international journal, I think it needs to be more broad. | Thank you for this comment.
Following the comment of the reviewer, we moved some part of the field (former lines 180-185) and microstructural observations (former lines 315-324; 377-381; 389-392) to the Supplementary Material S1. We also moved the paragraph 7.1 to the Suppl. Mat. S1 as suggested by the Reviewer#2 and re-numbered the other paragraphs accordingly. We have created a Reference list in the Suppl. Mat. where we added the refs. that are cited only here and not in the main text. |
| I think there needs to be some explicit statements in the introduction and conclusions about what new insights are provided by this study and how they are relevant to outstanding questions relating to the control of fluid flow by deformation bands. Are fundamental questions being answered? Are ideas proposed elsewhere given a more robust foundation? I think a focus on how the observations presented are relevant to, or support, proposed processes for cementation as discussed in section 7.3 would be good. But prior to all of this, there needs to be some discussion of what the outstanding questions are and why the reader should care about them. | -We think that in the introduction we explain the rationale of the paper (lines 74-104), what lacks in the existing literature (lines 133-154), and what is it the aim of this paper (lines 155-164). In the conclusions, and in particular in bullet points 2, 3, 6, 8 we sum up our insights.
-We have reorganized and rephrased the Introduction and, hopefully, now it is more in line with what asked by the reviewer.
-The discussions about how the observations support the mechanisms proposed in Section 7.2 are made in the Discussion section (7) and in particular in Sections 7.1 and 7.2. Our discussion is also supported by field, microstructural, and petrophysical data published in past works. |
| I think these changes would give the paper much more focus and attract the readers interest immediately. The observations in the paper should be limited to those that are relevant to these questions and or are genuinely new. The maps and thin section figures are impressive and the | Ok, thanks for the comment.

We have moved some field and microstructural observations, and Section 7.1 to the Supplementary Material S1 (see also response above in this file). We have reorganized and shortened other Sections as |

| | |
|---|---|
| descriptions complete and accurate but I found myself saying "What is new here? Is all this detail needed?" Many papers have contained such detailed observations of deformation bands and cements. What is essential to make the point? | well (see tracked changes).

-We hope the paper is more to the point now. |
| As explained below some of the conclusions are vague. What data could help strengthen them? Recommendations are made that other workers should use the types of data presented here. How? A discussion of these two points would also strengthen the paper in my opinion and give it a wider audience. | Do you mean the discussion section? ...in the Conclusions no recommendations were made. We think you refer to comments on lines 733 and 740 (below in this file).
Please, refer to the responses below.
See also edit in the text (lines 1215-1217 and 1222-1224) |
| As an overall recommendation I suggest accept with moderate to major revision. Perhaps resubmittal and another review is necessary if there are major changes. | |
| **Specific Comments** ||
| 50. "Fluid flow mechanisms...." I don't agree. The next 20 plus lines describe numerous studies addressing the effects of dbs on fluid flow, so it seems fairly well understood to me and heavily studied. And see the discussion and number of references listed in section 7.3. To justify the sentence in quotes I recommend stating clearly and explicitly exactly what is not currently understood and how the present study addresses and clarifies these problems. | Thanks for the comment.
Now the introduction has been rephrased. We explicitly state what is known and which are the open questions this paper is trying to address (lines 92-164). |
| Figures 1 and 2. Clarify what is meant by DB azimuth. Is this strike or dip direction? The azimuths range from +- 90 relative to what geographic direction? | Ok, we have done that (lines 291 and 306).
The azimuth data refers to the DBs strike and the $\pm$ 90° is relative to the north direction. |
| 171. I would find it much clearer if you would use the average strike or dip direction to refer to the DBs rather than EESSWWNNWWE. Sorry, that's what it looks like on the page! You have calculated the mean orientations of the distributions in Figs 1d and 2d, so you could use them in the text. | We have added the mean orientations as suggested by the reviewer (lines 354-355). We did the same for Bollène (see response to comment just below). However, we would like to keep the cardinal directions. |
| 244. Same comment as line 171. Reference to Figure 11b here should probably be 6b? | We have added the mean orientations as suggested by the reviewer (lines 445-447). See also the response just above. The reviewer is right about the fig. reference. We changed it accordingly. |
| 305. The microstructural observations that follow here are exhaustive. To my mind all of these features have been described elsewhere in studies of DBs. What is new here? What is relevant and essential to the main points and arguments of the paper? If it is not relevant then it can go in a supplement and or briefly summarized. It seems to me that many of the microstructural observations could be replaced by permeability data and other hydrogeologic data by these authors (from their other papers?) or summarized from the literature. Such data are much more relevant to the hydrogeological model proposed in Sections 7.3 and 7.4. | Thank you for this comment.
-Following the comment of the reviewer, we moved some part of the field (former lines 180-185) and microstructural observations (former lines 315-324; 377-381; 389-392) to the Supplementary Material S1.

-We have added a table in the Supplementary Material S3 where we summarize the petrophysical data for two field sites along with their references We reference to it in the text (lines 867, 921, 940). We have preferred to not add these data in the main text, because those data have been already published, and so that the paper does not become longer than it already is. |
| 591. Do you mean to say"...decrease of hydraulic | Ok corrected (lines 944-945). |

| | |
|---|---|
| conductivity dominates over the flow velocity increase caused by porosity reduction......." | |
| 624. I don't think hypotheses is the right word here. The field observations tend to confirm the theoretical flow simulations and experiments etc. I think this should be emphasized more in the paper overall and mentioned in the introduction. The three mechanisms described in this section have been invoked in other field studies and/or examined in the laboratory, and field observations in this study suggests that they are all relevant and or viable as possible explanations for the cement distributions. | We thank the reviewer for this comment. We rephrased the text (lines 984-986), however we would like to stress the following points: -As far as we know, the flow "slow down" has never been invoked as a possible cause for cement precipitation in DBs. -Regarding the role of "reactive microfractures and fine-grained comminution products" on cement precipitation, we refer to several papers that propose or mention this model as relevant to the precipitation of cements around DBs (lines 952-964). -We do not say that we are hypothesizing a cement precipitation induced by solute-sieving. We suggest the applicability of this model to cement precipitation associated with DBs, whereas this model was proposed via experiment with an analog for fault gauge (Whitworth et al., 1999). Why should this be emphasized in the introduction? We think that this is a sort of conclusive statement. In the introduction we state our aims (lines 155-164) and the 3 mechanisms are widely discussed after. |
| 634. it would be good to add a line or two somewhere about basic interpretation of cathodoluminescence colors for those who aren't familiar. | We have added a few lines in the methods where the controlling factors of CL characteristics (visual colors and intensity of emission) are summarized (lines 338-342). |
| 676. Add a reference to the measurements of DB hydraulic properties, or better , add the data to the paper as suggested above. | We have added the references as suggested by the reviewer (line 1098). -We have added a table in the Supplementary Material S3 where we summarize the petrophysical data for two field sites along with their references. We have preferred to not add these data in the main text, because those data have been already published, and so that the paper does not become longer than it already is. |
| 700. Can you expand on this? Are there any microstructural characteristics of the cement that would allow interpretation of its growth direction? | No, unfortunately we do not have enough microstructural evidences to assess the "net" growth direction of the cement. |
| 685 – 715. In general this discussion about the Bollene site seems reasonable but not definitive and rather underwhelming. E.g., Statements like " most likely" and "probably". Can you rework it to say what is known about the flow patterns and diagenesis definitively versus not. What about the three mechanisms described in section 7.3. Which do you think are applicable here (and at the Italy site), or would that just be total speculation? What additional info would be necessary to better understand the flow patterns and how they are controlled by the DBs and the resulting cement diagenesis. Ie, what could be a future research direction here? | -Ok, we rephrased the discussion about Bollène in Section 7.3 (lines 1106-1181) to make the discussion more solid. However, we would like to point out that these are discussions and, here, we make (working) hypothesis and propose the most robust (from our point of view) mechanisms/model considering all the data available (our and from literature). "In geology there are rarely any absolute models" (cit. Charrach, 2020 – JSG). -We discuss these mechanisms, as also their applicability to both field sites, in Section 7.2 (former Sect. 7.3) (lines 937-980), and Section 7.3 (former Sect. 7.4) (lines 1031-1032; 1083-1085; 1094-1095; 1164-1166). Certainly, in Bollène, the second and third mechanisms are not applicable since there is no |

| | spatial overlap (or it is rare) between DBs and cement. We state this also in the conclusions (bullet point 4). |
|---|---|
| | -About the future research direction... that is a good point. We answered to that in the conclusions (bullet point 7). Flow simulations and cement precipitation modeling could be used to reconstruct paleo-fluid flow pathways; further explore micro-scale fluid flow and diagenetic mechanisms that drove preferential cement precipitation along DBs; and constrain the reaction kinetics. These tools will also be helpful to validate the mechanisms involved in cement precipitation along DBs proposed in this work. |
| 726. I would say enhance rather than increase porosity reduction etc.... | Ok suggestion taken and implemented (line 1204). |
| 733. How would you include this information in a fault seal analysis. Give an example...it seems some sort of upscaling would need to be involved. E,g., Spatial density or proportion of cemented rock per unit area or length of fault? Something else? | Ok, we expanded on this point, and we added some references to studies in which DBs are incorporated into reservoir models and flow simulations (lines 1215-1218). |
| 740. Again, give an example. If someone is working with seismic data to do a fault seal analysis or reservoir engineering study how does the present study help them predict where the reservoir compartments are arranged etc. Elaborate. | Ok, we have elaborated about how this study could be helpful during reservoir characterization (lines 1222-1224). However, we would like to stress the following points: It could be difficult with standard seismic to observe the network of DBs and the associated cement nodules because in most of cases these features are below seismic resolution. That is why a field (analog) study, such as that presented in this work, is necessary during reservoir characterization since it allows to: (i) define the geometry and orientation distribution of DBs, (ii) understand which sets of DBs are associated with cement; (iii) evaluate the cement distribution and how it is arranged with respect to the DBs; (iv) evaluate how the compartments are arranged. Such information should be then integrated with other data, such as the density and clustering of DBs, the volume of cement along DBs, the petrophysical properties. This information can be then incorporated in reservoir (flow) models employed for hydrocarbon production planning. Anyway, all the attributes listed above are somewhat implicit in the reservoir characterization workflow, so we would like to not repeat them in the text. -Through the text, we point out that DBs and related cements (SDH) are subseismic features (lines 27,93 1209,1224, 1274), and that the study of an outcrop analog can improve the characterization of these features and in general be useful for the characterization of a faulted sandstone reservoir (lines 1217-1224). |
| 756. As noted above, I would say that the field observations support three mechanisms that have | The term "proposed" was substituted with "discussed" (Line 1250). |

| | |
|---|---|
| been proposed previously as relevant to the precipitation of cements around DBs. | -However, as stated above in this file to the comment at line 624, we would like to stress the following points:
-As far as we know, the flow "slow down" has never been invoked as a possible cause for cement precipitation in DBs.
-Regarding the role of "reactive microfractures and fine-grained comminution products" on cement precipitation, we refer to several papers that propose or mention this model as relevant to the precipitation of cements around DBs (lines 952-964).
-We do not say that we are hypothesizing a cement precipitation induced by solute-sieving. We suggest the applicability of this model to cement precipitation associated with DBs, whereas this model was proposed via experiment with an analog for fault gauge (Whitworth et al., 1999). |
| 763. Comment at line 685 in the text, apparently I missed this discussion in the text. Perhaps that section can be rewritten to more clearly state this. But then there should be some discussion as to why the difference between the applicable mechanisms at the two sites. Something about the host rocks, db microstructures, regional geology, regional flow patterns during cementation etc | -As stated above in this file to the comment at lines 685-715, we discuss these mechanisms, as also their applicability to both field sites, in Section 7.2 (former Sect. 7.3) (lines 937-980), and Section 7.3 (former Sect. 7.4) (lines 1031-1032; 1083-1085; 1094-1095; 1164-1166). Certainly, in Bollène, the second and third mechanisms are not applicable since there is no spatial overlap (or it is rare) between DBs and cement. We state this also in the conclusions (bullet point 4).

-The second and third mechanisms are not applicable to the Bollène site simply because the cement do not spatially overlap the DBs (e.g. lines 969-971).
-The difference in the cementation pattern between Loiano and Bollène could be related to several factors, such as the regional vs. local flow pattern and the hydrological conditions during cementation, to the fluid conditions and the rate of the process. To discuss the reason why this difference exists would be too speculative since, in this study, we did not cover all possible controlling aspects. |

---

## Author Comment (AC2) · 5 Aug 2020

**Response to James Evans (Reviewer #2) Comments**

**Ref: se-2020-81**

**Title: Structural control on fluid flow and shallow diagenesis: Insights from calcite cementation along deformation bands in porous sandstones**

**Journal: Solid Earth**

| Major comments | |
|---|---|
| **Comment** | **Response author** |
| I fill in the specific review criteria below, after some detailed comments on the text.

Overall this is a good contribution, with detailed observations of structures and cementation patterns. I suggest a slight recasting of the paper to the topic of the dbs and cementation, dropping a bit of the tectonic implications. I also have a few editorial changes that I will provide via written edits on the manuscript – sorry to go old school. | We appreciate very much the constructive comments and excellent suggestions of the reviewer that we think deeply improved the first version of our manuscript.

Suggested edits have been implemented and they are tracked in the revised manuscript.
Please, consider that the track-changes Word tool created some problems with line numbering and its jumping on the annotated manuscript. Line numbering of the "revised manuscript version with changes tracked" and "manuscript without tracked changes" may not coincide. The line numbering we use in this document (Author Response to Reviewer #2) refer to the revised manuscript with tracked change file.

We have moved some of the main text to the Suppl. Mat. without altering the main message of the paper. This is because both reviewers asked for keeping the focus on the paper on the DBs and their control on fluid flow and diagenesis. See also responses below. Also the introduction has been rephrased, and now it is more focused on DBs and cementation. |
| **Specific Comments** | |
| Lines 15- 20 can be edited a bit; edits attached. | Ok suggestion taken and implemented (lines 15-19). The term "string mapping" has been kept (line 19). The field map in Loiano has been produced by building a network of strings on the outcrop surface, that was used as a reference frame for detailed mapping. We do not intend "scanline mapping". |
| Line 33 – Wilson and Goodwin (2006) discuss these sorts of processes, as does Parnell et al, (2004 – j sed research); | The reviewer is right. These papers address important issues regarding the hydraulic behavior of DBs and their role on diagenesis. They have been referenced later in the Introduction, when we focus on DBs, as also in the discussions. (Wilson et al. 2006: lines 90, 146, 930, 1163; Parnell et al., 2004: lines 135, 146). When you mention "Wilson and Goodwin, 2006", maybe you mean "Wilson et al., 2006". Am I right? |
| Lines 53 -55; Min et al., 2001, and Shipton et al., 2002 provide data on db fault permeabilities; Petrie et al, 2013 show alteration / mineralization/ cementation patterns in db faults | Shipton et al. (2002) and Petrie et al., (2014) were added as suggested by the reviewer (lines 98-99). We have a reference to Main et al. at line 143. |
| Lines 50-85 – could this be trimmed somewhat? | We have rephrased the introduction. |

| | |
|---|---|
| Line 170-175, Figures 3 and 4. It is a statistically stronger way to examine the data as vector data; decimating into histograms is ok, but better to determine mean vector of the poles to the dbs, with dispersion statistics. | Do you mean Figs. 1 and 2? Am I right? We thank the reviewer for his comment. We replaced the frequency histogram of DBs azimuth with azimuth frequency rose diagrams (Figs 1d and 2d). This representation is similar to the one shown in Fig.1 in Del Sole et al. (2020 – MPG). Here, the stereoplot reports the best-fit gaussian curves for each DB's population found by the software (Daisy3; Salvini, 2004) and calculated from the corresponding frequency distribution (histogram data). The values reported on the right side of the azimuth frequency rose diagrams (Figs. 1d and 2d) are the mean orientation (strike; N±90°) and the standard deviation (±sd) for each population. |
| 186-187 – rephrase – cementation is a process, and you are describing here the distribution of zones, pods, etc., that reflect that cementation. | Ok corrected (lines 369-370). |
| Through the text, thee are sentences with phrases such as "more tabular", etc., but we don't always get what the more is relative to. | Ok, these sentences have been corrected. We hope the sentences are clear now. (lines 392-393 and Suppl. Mat. S1). |
| 307-308. Figure 9 doesn't really report porosity data; the data are in Del Sole and Antonellini, 2019; and the porosity data are derived from microscopy, so it's a bit of an indirect measure. | Ok corrected (line 517). The reviewer is right, these data are in Del Sole and Antonellini (2019). We think, however, that this has already been specified in the caption of Fig.9 (line 583). |
| 309 – oversized relative to what? | The term "oversize pores" indicates those (secondary) pores that are larger than the average pore size of primary porosity and also a larger diameter than that of adjacent grains (e.g. Schmidt et al., 1977 Bulletin of Canadian Petroleum Geology, 25(2), 271-290). E.g. oversized pores are due to dissolution of detrital grains (e.g. carbonate clast, feldspar). |
| 428-431, Figure 12. On figures like this, usually the negative numbers decrease upwards along the y axis...and I prefer the axes to be one the left and bottom like a normal graph - the origin will not be 0, 0 but that doesn't matter for delta values. And the text here can be tighted a bit, I think; | Thanks for this comment. We would like to keep the diagram of Fig. 12 as it is, following some of the most commonly used $\delta^{18}O$- $\delta^{13}C$ isotope diagram in literature (referenced in the manuscript: e.g. Hudson, 1977; Moore, 1989; Nelson and Smith, 1996; Pizzati et al., 2020). Regarding the diagram axis issue, we would like to keep them like this, since we are showing only the negative field/portion (-x, -y) of the diagram (references as above). The text in the legend was tightened as suggested by the reviewer. |
| Cement from the nodules of the Loiano samples have $\delta^{13}C$ values between -7.68 and -1.47 ‰ (V-PDB) and $\delta^{18}O$ values between -4.42 and -1.35 ‰ (V430 PDB) (Figure 12). The DBs-related nodules is characterized by isotope compositions between -5.41 and -1.47 ‰ (V-PDB) for $\delta^{13}C$, and between -4.42 and -1.40 ‰ for $\delta^{18}O$ (V-PDB). The bedding-parallel nodules has isotope compositions between -7.68 and -5.94 ‰ (VPDB) for $\delta^{13}C$, and between -2.09 and -1.35 ‰ (V-PDB) for $\delta^{18}O$. | We have revisited and shortened the text according to the reviewer suggestions typed here in the comment (lines 663-667). |
| 460-490 – I am not sure how much of this is needed. The focus of this paper is on the outcrop and microstructural observations and interpretations and | We did the edits suggested and moved the paragraph 7.1, along with the references therein, to the Suppl. Mat. S1 and re-numbered the other paragraphs in the |

| | |
|---|---|
| the regional tectonics do not seem to be the main point | main text accordingly. We created a Reference list in the Suppl. Mat. where we added the refs. that are cited only here and not in the main text. |
| 505 not clear what the meaning of the orange is here – do you mean orange cements? And orange color is redundant | Ok corrected. We hope it is clear now. (lines 788-789). |
| 510-515 – I am sorry – I got a little lost here with the terms like "likely preserved"; "suggest that cement dissolution.." "suggesting that cementation postdate..." There so many solid observations in this paper that I think you can state your interpreations more forcefully. | Thanks for this comment. Ok corrected. Now, the sentences you mention should be more solid (lines 788-794). |
| 519 and elsewhere – evidence is never plural evidences is not a word | Ok corrected along all the manuscript (lines 374, 545, 706, 801, 842, 1030) |
| 520. 800-1000 m | Ok suggestion taken and implemented (line 802) |
| 530-535 - are these bed parallel nodules basically related to diagenesis of some sort | If I understand well the reviewer point, yes. We think that bedding-parallel nodules are related to cement precipitation from flow focused along bedding, and in particular they are related to flow within sand levels (lines 810-837). In this Section (now Sect. 7.1) we discuss their occurrence, morphology, and a possible relationship with sandstone grain size. In Sect. 7.3 we also discuss the formation of bedding-parallel nodules considering the data available from this study and from literature. |
| 540-560 This is not very clearly written – please see written edits on the text | We implemented the suggested edits and rephrased the paragraph. We hope it is clear now. (lines 840-857). |
| 570- 715 – need to break this into paragraphs; Shorten, and see edits. I suggest separating the facts, and observations, and then discuss your interpretations; read through this section and eliminate all the clauses at the beginning of many of the sentences; In this section, look for all the 'coulds' in this section. My apologies – I think I am having COVID brain issues, but this section is pretty hard to read. I think it takes away from your work by not being a bit shorter, clearer, and organized. | The lines of this comment (570-715) refer to both Section 7.3 (now 7.2) and 7.4 (now 7.3). Considering the comments here at the left and the written edits on the manuscript, we believe, instead, that the reviewer refers only to Section 7.4 (now 7.3).

We accepted the edits suggested by the reviewer. We have broken this Section 7.3 in paragraphs, reorganized and shortened the text. When possible the "coulds" were replaced trying to make the discussion in this section more solid. |
| 593 594 – Precipitation occurs when the reaction is out of equilibrium. "slowing" flow could push fluids to be closer to equilibrium with the host rocks, thus, less likely to precipitate | -We thank the reviewer for the comment, this is an interesting point.
We added a possible explanation in text (lines 947-949). Below, we will discuss more about the reviewer point.
-At lower flow velocities the precipitation reaction has more time to proceed before the fluid leaves the system. We added 2 references where the effect of flow hydrodynamics on kinetics of precipitation was studied.
-We think that if fluids are enriched in constituents needed for calcite precipitation (Ca and bicarbonate), then the "equilibrium" you mention e.g. could just be reached through cement precipitation. We propose that a "slow down" of the flow could kinetically favor cement precipitation.
-As argued by the reviewer, at low flow velocity, the |

| | residence time could be long enough allowing the fluid to equilibrate with the host rock. In this case, the equilibrium between host rock and the percolating solution could be reached not far from the upstream side of the DB. This could support the asymmetric distribution of cement with respect to the DB, as observed in the field. |
| | –At higher flow velocity, there would be a higher and continuous supply of reagents, but the reaction would be disfavored. In fact, for most rock-forming minerals at ambient temperatures, chemical reactions at the solid surface are slow and thus rate limiting. If flow is fast, the reaction might not have time to occur. This is true if the fluid did not reach carbonate saturation (low saturation index). |
| | -In our case, it is also important to keep in mind, that the growth substrate is made mostly by quartz and feldspar, hence the mineralogy has a limited control over the process of mineral nucleation and growth [e.g. Pin and Singer, 2005, Geochimica et Cosmochimica Acta, 69(18), 4495–4504]. The precipitation of calcite over a silica substrate would need more time to occur when compared to a carbonate substratum (Stockman et al., 2014, Geochimica et Cosmochimica Acta,135, 231–250). |
| | -Another simple reason that supports our hypothesis is that nodules preferably occur along DBs where the porosity and permeability are lower than the surroundings and where the net flow is slower. Accordingly, the host rock is higher in porosity and permeability; here, the net flow is faster and the cement is absent or poor. This possibliy mean that the flow velocity has a role in the precipitation process, or at least to its initiation (See refs in line 949). |
| What aspect(s) of this work / interpretation(s) address, or apply to, db faults-fluid flow questions in general? | -Here, we bring some field evidence that show how DBs affect the flow pattern and control the distribution of diagenetic heterogeneities, i.e. nodules.
-We also argue about different potential mechanisms that drove cement precipitation within and around the DBs (Section 7.2). Most of the past field-based works were focused on the occurrence and the distribution of the cement, whereas cementation mechanisms received less attention.
-The mechanisms commonly employed to explain cementation within DBs (e.g. lines 141-147) are appliable and limited to specific conditions and need that the DB behaved as "conduits" (147-150). However, a lot of works demonstrate that commonly DBs reduce permeability and baffle the flow (lines 99-104).
-We discuss some mechanisms that are pertinent also in late-stage diagenesis (post-DBs formation) and in saturated conditions (phreatic environment), and that |

| | are of more general validity in low-permeability DBs Conclusions (bullet point 6). |
|---|---|
| Suggest moderate revisions. | |

---

## Editor Decision (ED1)

Dear Dr Del Sole,

Thank you for diligently addressing my previous comments on your manuscript. I find that this manuscript is much improved from the previous, particularly in the areas of readability/grammar and in providing a more nuanced discussion of data interpretation. There are, however, a few outstanding issues that need to be addressed. I would like to reconsider this manuscript following minor revisions. My hope is that this will be the last round and afterwards we can proceed to publication.

Below you will find several comments related to grammar and presentation. These are extremely minor, as your most recent revision made major advances in this area. The manuscript in its current form reads very well. More major concerns are present with respect to two areas of the presented science.

*"Slow Down" Effect:* The first, is that in your new discussion of the "slow down" effect, you mention that your inference is that the rate of the surface reaction is the rate-limiting step in cement accumulation. You then posited based on this inference that if flow is "fast", then the reaction "may not have time to occur". This is entirely counter to the classical geochemical view of rate-limiting reactions. Specifically, if the rate of the surface reaction is the rate-limiting step, then the overall rate of calcite growth is *invariant to transport rates*. See this image that I admittedly just pulled off of google, but it illustrates the point well.

[Figure]

If calcite growth is transport limited, then growth rates are proportional to the rates of transport. If it is surface reaction limited, then the rate of transport makes no difference. Thus, if you wish to continue to argue for the "slow down" effect as differentiated from solute sieving then that is no problem, but your explanation of it must be consistent with the broader body of knowledge

considering reactive transport. In short, I think your proposed mechanism needs a bit more thought.

*Oxygen Isotope Interpretations:* I appreciate your doing the calculation to estimate fluid d18O values, but in this version you simply mention that you did that and that they are consistent with your interpretations. The values need to be: 1) stated explicitly in the manuscript (rather than pointing the reader toward the Supplement); and 2) discussed relative to the *expected* values for meteoric fluids in the study area. The values you list in the supplement are a bit heavy for meteoric fluids - is this consistent with what has been observed for modern precipitation/ground water in the study area, or better yet, any values obtained from paleoclimatic studies?

See: *Giustini, Francesca, Mauro Brilli, and Antonio Patera. "Mapping oxygen stable isotopes of precipitation in Italy." Journal of Hydrology: Regional Studies 8 (2016): 162-181.*

**Line Referenced Comments**

89: As this sentence is currently constructed, an additional comma after "between" would be preferable.

92: Suggest rephrasing for english grammar clarity: "Our study also allows evaluation of the impact of…"

252: Thank you for adopting the more standard azimuth notation. Something has gone wrong here though - N334E?? That does not make sense. It should just be left at "334°". The N and E information are redundant. Please address here and throughout.

544-546: There is a problem with the logic here, in that a classical view of surface reaction limited growth implies that the overall growth rate is invariant with respect to the rate of transport.

555-560: The point I was making in my previous comments was that we very frequently see preferential calcite cementation of deformation bands that do not record significant cataclasis or fracturing. This implies that there must be something else (or at least something in addition to the role of fracturing) in helping to localize calcite cements in those structures. This makes them different from quartz cements, where the main thing seems to be fracturing. You do not need to change anything here. I bring this up only as an interesting point of discussion and potential impetus for future work.

585-590: I appreciate your adding this paragraph in response to my comments, but I do not think you actually need it. The other modifications you have made to this section make it a bit superfluous. You are welcome to keep it if you chose, but its elimination may make for more concise reading.

611-621: This still needs work. The opening sentence should more reasonably stated that "Oxygen isotope data also support a meteoric environment" or similar, and then go on to state the supporting data. The negative nature of the data say nothing on their own, and I also find the notion of any particular d13C value being indicative of meteoric environments preferentially questionable. You can certainly discuss aspects of the d13C signal that are *consistent with* a meteoric origin, but I do not think they can be an independent indicator. Oxygen, however, can. Toward that end, a bigger problem is that you mention calculating the fluid d18O, but do not discuss the inferred values here? Its a fundamental data type you argue to support your interpretation - it cannot be relegated to the supplemental information and not stated. The calculation can reasonably reside in the supplement, but you need to actually state the inferred fluid d18O values.

I see in the supplement that the values you calculate are between about -7 and -3. These actually strike me as slightly heavy for meteoric fluids, but those do vary by quite a lot over the surface of the earth. You will also need some brief discussion as to whether these values are consistent with meteoric fluids in the study areas, either now or from a paleoclimatic perspective around the inferred time of cement formation.

633: It is your inference that calcite precipitates where fluids slow down in the vicinity of DBs.

650-655: Same problem mentioned above.

731: Please plan to address issues with isotope interpretations here in the conclusions as well.

---

## Author Response (AR2)

Bologna (Italy), 07 September 2020

Ms. Ref. No.: se-2020-81

Title: Structural control on fluid flow and shallow diagenesis: Insights from calcite cementation along deformation bands in porous sandstones

Journal: Solid Earth

Dear Editor Randolph Williams,

We would like to submit our revised manuscript "Structural control on fluid flow and shallow diagenesis: Insights from calcite cementation along deformation bands in porous sandstones" by Leonardo Del Sole, Marco Antonellini, Roger Soliva, Gregory Ballas, Fabrizio Balsamo and Giulio Viola for consideration and eventual publication, as a research article, in the Special Issue in Solid Earth titled: "Faults, fractures, and fluid flow in the shallow crust".

We thank the Editor for his constructive comments and the help to improve our manuscript.

We hope that the explanations and the corrections made are satisfactory.

All authors have approved the revised manuscript and agree with its submission to Solid Earth.

Looking forward for your feedback,

Kind regards,

Leonardo Del Sole, Marco Antonellini, Roger Soliva, Gregory Ballas, Fabrizio Balsamo and Giulio Viola

**Response to Editor Comments**

**Ref: se-2020-81**

**Title: Structural control on fluid flow and shallow diagenesis: Insights from calcite cementation along deformation bands in porous sandstones**

**Journal: Solid Earth**

| Major comments | |
|---|---|
| **Comment** | **Response author** |
| Thank you for submitting this revised version and for diligently addressing reviewer comments. As you know, you received two reviews for this manuscript. The first review recommended consideration following major revisions. The second recommended consideration following only minor revisions. Having read the revised version and your responses to the two reviewers, I feel that significant progress was made toward eventual publication. I also feel, however, that additional improvements are required before the manuscript can move on to an accepted status. As such, I will reconsider your manuscript following additional minor revisions. | We appreciate very much the constructive comments of the Editor and the help to improve our paper. We hope to have addressed all issues raised in the review.

 Suggested edits have been implemented and they are tracked in the revised manuscript. Minor edits were also done to improve the grammar and fluency of the text.
 Please, consider that the line numbering of the "revised manuscript version with changes tracked" and "manuscript without tracked changes" may not coincide. The line numbering we use in this document (Author Response to Editor) refer to the revised manuscript with tracked change file. |
| Please address the following line referenced comments in a revised version of your manuscript. While completing those revisions, I request that you pay particular attention to comments regarding the scope of geochemical interpretations, and one of your preferred models for calcite precipitation in sandstones. | Please, refer to the following responses for detailed explanations and revisions made about the comments of the Editor. |
| I look forward to seeing a revised version of this work, and I believe it will ultimately make a good contribution to this special issue. | We thank the Editor for the comment. |
| **Specific Comments** | |
| 15: You only use "structural and diagenetic heterogeneities" twice in the abstract, so I suggest not defining the acronym here for the sake of text clarity. In fact, unless you are up against a space constraint I would personally recommend that you leave acronyms out of the abstract all together, but I will leave that to you to decide. | Ok, we removed the acronyms for "deformation band" (DB) and "structural and diagenetic heterogeneities" (SDH) from the abstract. We introduce these acronyms later in the Introduction (lines 53 and 66). |
| 22: The fact that isotope values are negative does not in itself support a meteoric environment, particularly when the scale of d18O (vpdb vs smow) has not been specified. Suggest just stating interpretation that isotopes support a meteoric fluid source and elaborate later. | Ok, corrected (line 22). |
| 38: Texture is a bit vague here. Please specify what is meant in this case. | -Ok, now we have specified what we mean by texture, and the term has been replaced with "textural characteristics" (line 54). We hope that now it is more clear.
 -In order to be more specific, with this term we mean that the development of features, such as DBs and nodules, changes the original structure and textural |

| | characteristics of the protolith (e.g. grain -size and -shape, grain orientation, packing density, relative proportion of grains, mineral content, matrix material and type, cement type and degree of cementation, grain contact relationships). |
|---|---|
| 47: The paper by Williams et al. you are citing was actually published in 2016, not 2017. | Ok, the publication year was corrected (lines 63 and 1280-1282). |
| 50: Please remove the word "different", it is not necessary. | Ok, done (line 67). |
| 67: I do not understand the use of "besides" here. Perhaps it should be "in particular" to draw specific attention to calcite of all the cements that can form in DB's? | The Editor is right. Suggestion taken and implemented (line 99). |
| 68: After much thought, I agree with Reviewer #1 - this statement under emphasizes the advances that have been made in this field. Please be more specific about what aspects remain poorly constrained. I think it would be fair to say that the controls on the origin and distribution of calcite cements remain poorly constrained, both in DB's or even in undeformed reservoir rocks. | Ok, we have modified the sentence (lines 100-102), however we would like to stress the following point: -the comment of the Reviewer #1 to which the Editor refers is "*50. "Fluid flow mechanisms...." I don't agree. The next 20 plus lines describe numerous studies addressing the effects of dbs on fluid flow, so it seems fairly well understood to me and heavily studied. ...*", and it refers to this sentence "*Fluid-flow mechanisms and evolution within DBs, remain also poorly understood particularly with regards to diagenetic processes*" (former lines 49-50: manuscript vers.1). That part has been already rephrased in the previous revised version of the manuscript as follows "*Diagenetic processes related to fluid-flow mechanisms and evolution within DBs are not fully constrained...*" (lines 68-69: revised manuscript; lines 100-101: current revised version). Following the Editor's comment, we have added this to the previous sentence: "...and in particular, how DBs steer the origin and distribution of calcite cement remain poorly understood" (lines 101-102). We are aware that different processes have been proposed to explain cement precipitation in DBs (lines 102-108: current revised version) but we argue that "*these mechanisms appear to be limited to specific conditions...assume that DBs behaved as fluid conduits...etc.*" (lines 108-112: current revised version) and they may not explain the wide range of occurrences of DBs associated with cement. In this view, we think that it is fair to say that diagenetic mechanisms associated to DBs are not fully constrained. Please, see also the response to the comment to line 80 (below in this file) |
| 70: Wilson et al. (2003, Geology) would be another appropriate reference for vadose zone effects in DB's. | Ok, reference added (lines 103 and 1283-1285). |
| 80: I think this line needs a bit more specificity. A lot of work has actually looked at and thought about why low-permeability DB's are selectively cemented by quartz (authors Laubach, Millikin, Williams, Lander, etc). With respect to calcite, I think your statement is true, so maybe that is the specificity that needs added? | The sentence was rephrased to better clarify what is known and what is lacking (lines 112-115); however, we would like to stress the following point. We are aware that several papers looked at and thought about why low-permeability DB's are selectively cemented by quartz. However, as an example, the paper from Milliken et al. (2005 – |

| | AAPG) describe "*preferential early cementation in the bands...*". The paper from Milliken and Reed (2002 – Gulf Coast Association of Geological Societies Transactions, Vol. 52) states "*...bands record a history of progressive development that partially overlaps the timing of quartz cement emplacement in the sandstones*". We clearly state that we are looking for cementation mechanisms that account for "post-deformation" cement precipitation, and not limited to cement precipitation in the early stage of DB formation. The paper by Williams et al. (2015) has been already cited and now we have added also the work by Lander et al. (2009) (line 115). |
|---|---|
| 83: This opener sentence needs revised for clarity. It is difficult to follow. It is also an important moment to frame your study for the reader. | The opener sentence of this paragraph was revised (lines 117-118). We hope that now it is more clear. |
| 84: Many would be willing to debate about how novel it is to take a cross-disciplinary approach such as this to look and diagenesis and fluid flow in deformation bands. Stated as generally as it is, many of us have done very similar. If this line is kept, I suggest merging it with the following which is much more specific. | We have revised and merged the two sentences as suggested (lines 118-121). |
| 101: This information needs a citation. | Ok, reference added (line 154). |
| 116: Please cite this information. | Ok, reference added (line 170). |
| 118: Do you really know this value to the second decimal place? Sort of precludes any variability at the outcrop scale. | Ok, we removed the decimal numbers (line 172). However, please consider that the previous value (22.05) is not an average of different measurements, but a single porosity measurement result measured by mercury injection porosimetry at the study site (Ballas et al., 2014). The porosity range of Turonian Sandstones in the area is also reported in the text. |
| Figure 1: Please revisit this figure in terms of text size. Some of this is too small to be easily read. As a general rule, the minimum font size that should be used is 8-point. | The font size has been increased as much as possible without changing the panels organization. We hope that know it is more readable. |
| Figure 2: Same comment as above. | Same response as above. |
| 145: Please pluralize "altitude". | Ok, done (line 204). |
| 150: To my mind "orientation" encompasses both strike and dip so "dip" here is redundant. | The term "dip" was removed (line 209). |
| 154: Natural light? As in an old microscope that actually reflects sunlight? | The Editor is right. The term was corrected (line 213). |
| 156-157: Texture is used twice here in two different contexts. Please specify. The word is a bit ambiguous and is being used a lot in this paragraph. | We have added some textural characteristics for clarity (lines 216-217). In order to be more clear, we would like to point out that the term is referred to the textural characteristics of the constituent components relative to the host rock and DBs (e.g. grain or particle size, shape, appearance, sorting, and arrangements and also the grain contact relationships) and cement (e.g. cement type and degree of cementation, cement-crystals size, shape, arrangements, and grain–cement bounding). |
| 178: "Prediction uncertainty"? | We removed the term "prediction" (line 243). We apologize for the typo. |

| | |
|---|---|
| 181-182: Please but the DB orientations in standard azimuthal form here and throughout (i.e. 340 instead of N20W). It is considerably easier to follow and in my opinion is generally the standard for structural work of this type. | Suggestion taken and implemented (lines 248 and 318-320). |
| 183: Suggest deleting "mostly". | We would like to keep the term "mostly" since some DBs dip also in east and north quadrants (Fig.1d). |
| 217-218: Concretions/nodules occurring preferentially in coarser-grained horizons is consistent with observations made by Hall et al (2004, J. Sed. Res.) and Davis et al (2006, Sed. Geol.). | Yes, the effect of grain size and permeability variations on nodules development has been discussed in Section 7.1 and here we refer to the citations suggested by the Editor (lines 556). |
| Figure 4: What is the red box in panel C? | It indicates the figure in panel (f). |
| 450-455: Isn't is also true that you did not observe any crushed or comminuted calcite in the db's? It seems that would be the most compelling evidence that DB formation preceded calcite cementation? | The Editor is right, however, the discussion in lines 520-532 is referred to "macroscopic" evidence that DB formation preceded calcite cementation. The "microscale" evidence to which the Editor refers is discussed just below (lines 538-547) and, in particular, in lines 538-539 and 544-547. |
| 459: The last sentence does not follow from the previous. You confirmed that cement precipitation postdates db formation based on your observations. The following discussion about why that might be the case (strengthening) is appropriate, but it does not provide any additional confirmation for your original interpretation. | We have kept the discussion about the strengthening (lines 532-537) and we have removed the last sentence to which the Editor refers, to make clearer that this discussion serves only as a supporting argument.

However, we would like to stress the following point: the fact that the presence of cement would have prevented DBs development and favored the formation of joints is a supporting argument that the sandstones were not cemented prior to DB formation. DBs are the main structural element we observe, and a few later joints develop after DBs and nodule formation, and joints crosscut them (lines 292-293). |
| 470: Isn't this sort of a truism in describing them as "db-parallel" nodules? | The Editor is right, but we state this (line 548) to highlight that no other apparent factor that control the occurrence and distribution of nodules associated with DB has been observed, whereas bedding-parallel nodules occurrence and location appear to be controlled mainly by grain-size and contrast in grain-size within the host rock, and therefore by host rock permeability (lines 549-556). |
| 497: Please change "poral" to simply "pore". | Suggestion accepted (line 582). |
| 512: Please just state simply whether you infer that cataclasis increases or decreases tortuosity | Ok, done (line 597). |
| 528: I think the term would be "transmissivity" and I would argue it is not appropriate for a comparison here because it inherently includes a component of element thickness. Maybe just hydraulic conductivity would be preferable? | We agree with the Editor. The term was changed in "hydraulic conductivity" (line 613). |
| 527-540: This mechanism needs more discussion. First, whether or not an increase in residence time would lead to increased precipitation depends greatly on the details of the local geochemistry. I would argue that the primary control on calcite precipitation is the degree of supersaturation in the fluid. Is there an inherent reason that a simple increase in residence | -We thank the Editor for the comment, this is an interesting point. We have expanded the discussion to cover this point (lines 624-629 and 665-677). Below, we will discuss more about the Editor point. -The Editor's observations are correct: we need to know the geochemical conditions of the fluids and the reaction kinetics to be able to assess the processes |

time would alter supersaturation with respect to calcite? One could actually argue the reverse - an increase in residence time means that for every mol of calcite precipitated we see a corresponding decrease in the degree of supersaturation, and decreasing rates of precipitation. Higher flow rates could more effectively maintain optimal supersaturations - at high flow rates, you are always bringing in a fresh source of supersaturated fluids. Also, this mechanism is not so easily distinguished from that proposed by Whitworth. I understand the title and abstract of that work describe "gouge", but in reality what they were looking at was pulverized sand. There isn't really an inherent difference between that and what is typically observed in deformation band faults.

of cement precipitation in the band and its evolution through time. This, however, is very complex, to assess. Also, we believe that the supersaturation conditions of the source fluids would change over time (limited availability of carbonate to dissolve in the sandstone framework <4%). For this reason, a coupled fluid flow reactive model is necessary to evaluate the effects of the flow and that of the geochemical conditions; this is what we are working on right now.

-What the Editor points out is right if the supersaturated fluid comes from a continuous source with a constant degree of supersaturation. This is unlikely in our case, given the limited amount of carbonate available for dissolution in the sandstone framework.

-Having said that, a first simple observation that supports our hypothesis is that cement precipitation occurs literally where the flow slows down, i.e. the DBs. Nodules are found along DBs where the hydraulic conductivity and permeability are lower than the surroundings and where the net flow is slower (lines 617-622; see also Del Sole and Antonellini, 2019). This suggests that flow velocity has a role in the precipitation process, or at least to its initiation (See also refs in line 624).

-A "slow down" of the flow could kinetically favor cement precipitation. In fact, for most rock-forming minerals at ambient temperatures, chemical reactions at the solid surface are slow and thus rate limiting. If flow is fast, the reaction might not have time to occur. At lower flow velocities the precipitation reaction has more time to proceed before the fluid leaves the system. We cite 2 references where the effect of flow hydrodynamics on kinetics of precipitation was studied (line 624).

-Regarding the characteristics of the growth substrate, in our case the substrate (sandstones grain) is distinct from the cement (calcite). The precipitation of calcite over a non-calcite-based material (e.g. silica substrate) would require more time (Stockman et al., 2014, Geochimica et Cosmochimica Acta,135, 231–250) to occur (i.e. to lower the energy barrier for nucleation) when compared to a carbonate substratum.

-I will try to answer to my best the Editor comment about the similarity of the "slow down" with the model proposed by Whitworth et al. (1999).
Below, we will give our arguments that these two mechanisms are similar but are not the same. See also lines 665-671 in the manuscript.
-First of all, the "slow down" effect does not consider a solute-sieving or "filter" effect that would allow the physical accumulation/concentration of the solute on

| | the high-pressure side of the membrane and aid calcite precipitation. In this view, the "solute-sieving" model could explain well the asymmetric distribution of cement with respect to the DB, whereas it can hardly explain the occurrence of symmetric-distributed cement nodules around DBs. Moreover, the "solute-sieving" model does not take into consideration the effect of the "membrane" on the flow velocity, and the solution is forced through the "membrane" at a constant solution flux/flow rate. The "slow down" effect, instead, is based on the effect of the DB on the net flow, and several papers demonstrated that the DBs can strongly influence the fluid velocity field. For these reasons we think that both models, which are similar, could have worked to trigger cement precipitation and nodule formation. |
|---|---|
| 541-545: The citations that showed increase rates of cementation in deformation bands due to surface chemistry effects associated with fracturing are all specific to quartz cements and I think that should be specified. The hypothesis has many problems when applied to calcite cements in sandstones, in part because the cementing phase is distinct from the substrate (grains), and in part because deformation bands that exhibit little to no cataclastic deformation (i.e.. Work by Williams, Mozley, Goodwin, Rawling, etc) often also show preferential cementation. A bit more discussion could be added here covering those points. | We have expanded the discussion to cover this point (lines 640-644), however, we would like to stress the following points:
 -There are other papers that describe calcite cement associated with cataclastic DBs and that mention the effect of crushed grains and tiny pores as sites of cement precipitation (e.g. Antonellini et al., 1994 – JSG; Pizzati et al., 2019 – GSA) (lines 645-647).
 -We are not sure to understand the "problem" in the second point raised by the Editor. There are several papers describing preferential (quartz or calcite) growth in DBs with cataclasis because of the high concentration of nucleation sites on the newly created fracture surfaces (e.g. Antonellini, et al., 1994 – JSG; Knipe et al., 1997; Fisher and Knipe, 1998; Philit et al., 2015). They are referenced in lines 645-647. If this mechanism is effective in DBs that exhibit little to no cataclastic deformation, it should be even more efficient with newly formed microfractures in cataclastic DBs. Am I wrong? |
| 575: Please avoid 1-sentence paragraphs. | Ok, we have merged this paragraph with the next (lines 687-689). |
| 579-590: With respect, I do not see how any of these factors point necessarily to a meteoric phreatic environment. I have seen literally every one of these features in hydrothermal calcite deposits formed at several kilometers depth, where highly evolved basinal fluids were responsible for cement formation. CL color is a particularly bad metric for making these conclusions. Drusy textures certainly indicate a phreatic environment, but they have nothing to do with whether the fluid was meteoric. The only reliable indicator of meteoric fluids present in this study is the oxygen isotope data, and those data are reasonably compelling. Moreover, the maximum burial depth of these units is 1 km. Why not just focus on those more obvious sources of justification for a meteoric environment? | Ok, we have removed the indicators that are poorly reliable for this discussion, as suggested by the Editor (lines 690-697). Then, we focus on the oxygen isotope data and the burial depth to argue about the environment of precipitation and the possible fluid source (lines 698-734).

 However, we would like to point out that no evidence of alteration and mineralization (other than calcite) related to hydrothermal activity has been reported, neither in our study site nor in the surrounding area. Moreover, I suppose that if that calcite was of hydrothermal origin the oxygen isotope composition would be different from what we observe (Fig. 12a). I would expect more enriched and positive values for $_{18}O$ in hydrothermal calcite. |
| 593-594: The negative nature of the isotope values alone does not support this interpretation. Stable | Thanks for this comment. We have added a paragraph in the Supplementary Material (S4) where |

| | |
|---|---|
| isotope values mean very little without some inference on temperature and therefore inferred fluid compositions. In this case, your maximum burial depths are less than 1 km. This should allow you to assume a reasonable temperature range and calculate potential fluid d18O compositions. Without that, the interpretations related to meteoric environments remain uncertain. | we show $d_{18}O$ of parent fluids calculated with the Friedman and O'Neal (1977) formula, both for Loiano and Bollène. Please, see also lines 699-700. |
| 606: I am not sure why you are trying to pile up so much evidence that the system was fluid saturated during cement formation? There is no evidence of vadose cementation (e.g. pendant or meniscus cement geometries, micritic calcites, etc) and plenty of evidence for phreatic (sparry, drusy calcite, poikilotopic textures). Why pull in other considerations which are quite frankly a bit tenuous in their relationship to phreatic environments specifically? | Ok, we have removed this part (former start line 736). |
| 632-645: See comments above regarding what CL and isotopic variations reasonably can and cannot say without additional constraints. The degree to which CL variations are being used to infer fluid chemistry variations here is questionable. Both the CL activator (Mn) and quencher (Fe) are redox and pH dependent to varying degrees. How then do we so directly link CL colors to redox?? | The discussion about the zonation of CL colors has been removed. We have also added a paragraph in the Supplementary Material (S4) where we show $d_{18}O$ of parent fluids calculated with the Friedman and O'Neal (1977) formula Please, see also lines 822-823. We would like to keep the interpretation that dark luminescence could suggest an oxidizing environment, and thus a meteoric environment. Moreover, as suggested by the Editor in the previous comment, poikilotopic textures could suggest a phreatic environment. Therefore, using both the indicators (texture and luminescence), we think that it is possible to suggest a meteoric phreatic environment of precipitation for nodules in Bollène. This is just a hypothesis. Then, our interpretation is supported by the oxygen isotope data, the burial depth (lines 821-825), and the presence of the massive calcrete (lines 825-827). |
| 680-681: But how would we include such information? What is the spatial distribution of these features? How large are they on average? Perhaps suggest some future research imperatives along these lines? | How we would include this information has been already answered in lines 873-875. The spatial distribution, thickness and other properties of DBs and nodules have been described in the Sections 4.1 (Loiano) and 4.2 (Bollène) and in the field maps of figures 3 and 6. These data and petrophysical and mechanical properties of these features are also reported in past published papers (e.g. Del Sole and Antonellini, 2019; Del Sole et al., 2020 for Loiano; Ballas, 2013; Ballas et al., 2014 for Bollène). I am not sure why should these properties be presented here in the discussions. This is just a general argument to stress that this type of features should be considered during reservoir characterization in porous sandstones. Some future research directions are suggested in lines 675-677 and in the conclusions, bullet point (7). |
| 707-710: See comments above. Fluid "slow down" is not so easily differentiated from the solute sieving | -I will try to answer to my best the Editor comment about the similarity of the "slow down" with the |

| | |
|---|---|
| hypothesis. | model proposed by Whitworth et al. (1999). Below, we will give our arguments that these two mechanisms are similar but are not the same. See also lines 665-671 in the manuscript.

-First of all, the "slow down" effect does not consider a solute-sieving or "filter" effect that would allow the physical accumulation/concentration of the solute on the high-pressure side of the membrane and aid calcite precipitation. In this view, the "solute-sieving" model could explain well the asymmetric distribution of cement with respect to the DB, whereas it can hardly explain the occurrence of symmetric-distributed cement nodules around DBs. Moreover, the "solute-sieving" model does not take into consideration the effect of the "membrane" on the flow velocity, and the solution is forced through the "membrane" at a constant solution flux/flow rate. The "slow down" effect, instead, is based on the effect of the DB on the net flow, and several papers demonstrated that the DBs can strongly influence the fluid velocity field. For these reasons we think that both models, which are similar, could have worked to trigger cement precipitation and nodule formation. |
| 718: See comments above re: interpretation of isotopic results. | We have added a paragraph in the Supplementary Material (S4) where we show $d_{18}O$ of parent fluids calculated with the Friedman and O'Neal (1977) formula, both for Loiano and Bollène. Thus, we hope it is ok if we keep the sentence as it is. |

[revised manuscript text omitted]

ha eliminato: s

ha eliminato: meteoric

ha eliminato: with reducing conditions (low pO₂)

ha eliminato: Hiatt and Pufhal, 2014;

ha eliminato: that occur

ha eliminato: ,

ha eliminato: consists of

ha eliminato: meteoric

ha eliminato: In contrast, the dominant bright cathodoluminescence response suggests an environment with reducing (low pO₂) geochemical conditions. Cement zonation could be triggered by the shifting from more reducing (bright CL) to more oxidizing (dark-CL) conditions and vice versa (Barnaby and Rimstidt, 1989) due to water table fluctuations (Li et al., 2017). …

[revised manuscript text omitted]

---

## Author Response (AR3)

Ms. Ref. No.: se-2020-81

Title: Structural control on fluid flow and shallow diagenesis: Insights from calcite cementation along deformation bands in porous sandstones

Journal: Solid Earth

Dear Editor Randolph Williams,

We would like to submit our revised manuscript "Structural control on fluid flow and shallow diagenesis: Insights from calcite cementation along deformation bands in porous sandstones" by Leonardo Del Sole, Marco Antonellini, Roger Soliva, Gregory Ballas, Fabrizio Balsamo and Giulio Viola for consideration and eventual publication, as a research article, in the Special Issue in Solid Earth titled: "Faults, fractures, and fluid flow in the shallow crust".

We thank the Editor for his constructive comments and the help to improve our manuscript. We hope to have addressed all issues raised in the review.

All authors have approved the revised manuscript and agree with its submission to Solid Earth.

Looking forward for your feedback,

Kind regards,

Leonardo Del Sole, Marco Antonellini, Roger Soliva, Gregory Ballas, Fabrizio Balsamo and Giulio Viola

**Response to Editor Comments**

Ref: se-2020-81

**Title: Structural control on fluid flow and shallow diagenesis: Insights from calcite cementation along deformation bands in porous sandstones**

**Journal: Solid Earth**

| CommentResponse authorDear Dr Del Sole, Thank you for diligently
addressing my previous comments on your
manuscript. I find that this manuscript is much
improved from the previous, particularly in the areas
of readability/grammar and in providing a more
nuanced discussion of data interpretation. There are,
however, a few outstanding issues that need to be
addressed. I would like to reconsider this manuscript
following minor revisions. My hope is that this will
be the last round and afterwards we can proceed to
publication.Suggested edits have been implemented and they are
tracked in the revised manuscript version with changes tracked"
and "manuscript without tracked changes" may not
coincide. The line numbering we use in this
document ( Author Response to Editor V2 ) refers to
the revised manuscript with tracked change file.Below you will find several comments related to
grammar and presentation. These are extremely
minor, as your most recent revision made major
advances in this area. The manuscript in it surrent
form reads very well. More major concerns are
present with respect to two areas of the presented
science.After some thought we decided to remove the "slow
down" effect you mention
that your inference is that the reat of the surface
reaction is the rate-limiting step in cement
accumulation. You then posited based on this
inference that flow is "fast", then the reaction
"may not have time to occum". This is entirely
counter to the classical goochemical view of rate-
limiting reactions. Specifically, if the rate of transport
rates. See this image that I admittedly just pulled off
of google, but it illustrates the point well.After some thought we decided to remove the "slow
down" effect as differentiated
from solute science in the strate of transport
makes no difference. Thus, if you wi | Major comments                                             |                                                              |  |  |  |
|-------------------------------------------------------------------------------------------------------------------------------------------------------------------------------------------------------------------------------------------------------------------------------------------------------------------------------------------------------------------------------------------------------------------------------------------------------------------------------------------------------------------------------------------------------------------------------------------------------------------------------------------------------------------------------------------------------------------------------------------------------------------------------------------------------------------------------------------------------------------------------------------------------------------------------------------------------------------------------------------------------------------------------------------------------------------------------------------------------------------------------------------------------------------------------------------------------------------------------------------------------------------------------------------------------------------------------------------------------------------------------------------------------------------------------------------------------------------------------------------------------------------------------------------------------------------------------------------------------------------------------------------------------------------------------------------------------------------------------------------------------------------------------------------------------------------------------------------------------------------------------------------------------------------------------------------------------------------------------------------------------------|------------------------------------------------------------|--------------------------------------------------------------|--|--|--|
| Dear Dr Del Sole, Thank you for diligently
addressing my previous comments on your
manuscript. I find that this manuscript is much
improved from the previous, particularly in the areas
of readability/grammar and in providing a more
nuanced discussion of data interpretation. There are,
however, a few outstanding issues that need to be
addressed. I would like to reconsider this manuscript
following minor revisions. My hope is that this will
be the last round and afterwards we can proceed to
publication.
Below you will find several comments related to
grammar and presentation. These are extremely
minor, as your most recent revision made major
advances in this area. The manuscript in its current
form reads very well. More major concerns are
present with respect to two areas of the presented
science.
"Slow Down" Effect: The first, is that in your new
discussion of the "slow down" effect, you mention
that your inference is that the rate of the surface
reaction is the rate-limiting step in ement
accumulation. You then posited based on this
inference that if flow is "fast", then the reaction
"may not have time to occur". This is entirely
counter to the classical geochemical view of rate-
limiting reactions. Specifically, if the rate of the
surface reaction limited, then growth
rates. See this image that I admittedly just pulled off
of google, but it illustrates the point well.
If calcite growth is transport limited, then growth
rates are proportional to the rates of transport
rakes no difference. Thus, if you wish to continue to
argue for the "slow down" effect as differentiated
from solute seiving then that is no problem, but your
explanation of it must be consistent with the brooder                                                                                                                          | Comment                                                    | Response author                                              |  |  |  |
| addressing my previous comments on your
manuscript. I find that this manuscript is much
improved from the previous, particularly in the areas
of readability/grammar and in providing a more
nuanced discussion of data interpretation. There are,
however, a few outstanding issues that need to be
addressed. I would like to reconsider this manuscript
following minor revisions. My hope is that this will
be the last round and afterwards we can proceed to
publication.
Below you will find several comments related to
grammar and presentation. These are extremely
minor, as your most recent revision made major
davances in this area. The manuscript in its current
form reads very well. More major concerns are
present with respect to two areas of the presented
science.
"Slow Down" Effect: The first, is that in your new
discussion of the "slow down" effect, you mention
that your inference is that the rate of the surface
reaction is the rate-limiting step, then the
overall rate of calcite growth is invariant to transport
rates. See this image that I admittedly just pulled off
of google, but it illustrates the point well.
If calcite growth is transport limited, then growth
rates are proportional to the rates of transport. If it is
surface reaction limited, then growth
rates are proportional to the rate of thrasport
makes no difference. Thus, if you wish to continue to
argue for the "slow down" effect as differentiated
from solute seiving then that is no problem, but your
explanation of it must be consistent with the broader                                                                                                                                                                                                                                                                                                                     | Dear Dr Del Sole, Thank you for diligently                 | We appreciate very much the constructive comments            |  |  |  |
|  <li>manuscript I find that this manuscript is much
improved from the previous, particularly in the areas
of readability/grammar and in providing a more
nuanced discussion of data interpretation. There are,
however, a few outstanding issues that need to be
addressed. I would like to reconsider this manuscript
following minor revisions. My hope is that this will
be the last round and afterwards we can proceed to
publication.</li> <li>Below you will find several comments related to
grammar and presentation. These are extremely
minor, as your most recent revision made major
advances in this area. The manuscript in its current
form reads very well. More major concerns are
present with respect to two areas of the presented
science.</li> <li>"Slow Down" Effect: The first, is that in your new
discussion of the "slow down" effect, you mention
that your inference is that the rate of the surface
reaction is the rate-limiting step, then the
overall rate of calcite growth is invariant to transport
rates. See this image that I admittedly just pulled off
of google, but it illustrates the point well.</li> <li>If calcite growth is transport limited, then growth
rates are proportional to the rates of transport
makes no difference. Thus, if you wish to continue to
argue for the "slow down" effect as differentiated
from solute sieving then that is no problem, but your
explanation of it must be consistent with the broader</li>                                                                                                                                                                                                                                                                                                                                                                                                         | addressing my previous comments on your                    | of the Editor and the help to improve our paper.             |  |  |  |
|  <li>improved from the previous, particularly in the areas of readability/grammar and in providing a more nuanced discussion of data interpretation. There are, however, a few outstanding issues that need to be addressed. I would like to reconsider this manuscript following minor revisios. My hope is that this will be the last round and afterwards we can proceed to publication.</li> <li>Below you will find several comments related to grammar and presentation. These are extremely minor, as your most recent revision made major advances in this area. The manuscript in its current form reads very well. More major concerns are present with respect to two areas of the presented science.</li> <li>"Slow Down" Effect: The first, is that in your new discussion of the "slow down" effect, you mention that your inference is that the rate of the surface reaction is the rate-limiting step, in cement accumulation. You then posited based on this inference that if flow is "fast", then the reaction "may not have time to occur". This is entriely counter to the classical geochemical view of rate-limiting reactions. Specifically, if the rate of the surface reaction limited, then growth rates are proportional to the rates of transport. If it is surface reaction limited, then growth rates are proportional to the rates of transport anakes no difference. Thus, if you wish to continue to argue for the "slow down" effect as differentiated for mosolute sivening then that is no problem, buy your explanation of it must be consistent with the broader</li>                                                                                                                                                                                                                                                                                                                                                                                       | manuscript. I find that this manuscript is much            | We hope to have addressed all issues raised in the           |  |  |  |
| of readability/grammar and in providing a more
nuanced discussion of data interpretation. There are,
however, a few outstanding issues that need to be
addressed. I would like to reconsider this manuscript
following minor revisions. My hope is that this will
be the last round and afterwards we can proceed to
publication.
Below you will find several comments related to
grammar and presentation. These are extremely
minor, as your most recent revision made major
advances in this area. The manuscript in its current
form reads very well. More major concerns are
present with respect to two areas of the presented
science.
"Slow Down" Effect: The first, is that in your new
discussion of the "slow down" effect, you mention
that your inference is that the rate of the surface
reaction is the rate-limiting step in cement
accumulation. You then posited based on this
inference that if flow is "fast", then the reaction
"may not have time to occur". This is entirely
counter to the classical geochemical view of rate-
limiting reactions. Specifically, if the rate of the
surface reaction is the rate-limiting step, then the
overall rate of calcite growth is invariant to transport
rates. See this image that 1 admittedly just pulled off
of google, but it illustrates the point well.
If calcite growth is transport limited, then growth
rates are proportional to the rates of transport
makes no difference. Thus, if you wish to continue to
argue for the "slow down" effect as differentiated
from solute seiving then that is no problem, but your
explanation of it must be consistent with the broader                                                                                                                                                                                                                                                   | improved from the previous, particularly in the areas      | review.                                                      |  |  |  |
| nuanced discussion of data interpretation. There are,
however, a few outstanding issues that need to be
addressed. I would like to reconsider this manuscript
following minor revisions. My hope is that this will
be the last round and afterwards we can proceed to
publication. Here is the tracked in the revised manuscript with changes tracked"
and "manuscript without tracked changes" may not
coincide. The line numbering we use in this
document ( Author Response to Editor V2 ) refers to
the revised manuscript with tracked change file.
Below you will find several comments related to
grammar and presentation. These are extremely
minor, as your most recent revision made major
advances in this area. The manuscript in its current
form reads very well. More major concerns are
present with respect to two areas of the presented
science.
"Slow Down" Effect: The first, is that in your new
discussion of the "slow down" effect, you mention
that your inference is that the rate of the surface
reaction is the rate-limiting step in cement
accumulation. You then posited based on this
inference that if flow is "fast", then the reaction
"may not have time to occur". This is entirely
counter to the classical geochemical view of rate-
limiting reactions. Specifically, if the rate of the
surface reaction is the rate-limiting step, then the
overall rate of calcite growth is invariant to transport
rates. See this image that I admittedly iust pulled off
of google, but it illustrates the point well.
If calcite growth is transport limited, then growth
rates are proportional to the rates of transport
makes no difference. Thus, if you wish to continue to
argue for the "slow down" effect as differentiated
from solute sieving then that is no problem, but your
explanation of it must be consistent with the broader            | of readability/grammar and in providing a more             | Suggested edits have been implemented and they are           |  |  |  |
| however, a few outstanding issues that need to be
addressed. I would like to reconsider this manuscript
following minor revisions. My hope is that this will
be the last round and afterwards we can proceed to
publication.Please, consider that the line numbering of the
"revised manuscript without tracked changes" may not
coincide. The line numbering we use in this
document ( Author Response to Editor V2 ) refers to
the revised manuscript with tracked change file.Below you will find several comments related to
grammar and presentation. These are extremely
minor, as your most recent revision made major
advances in this area. The manuscript in its current
form reads very well. More major concents are
present with respect to two areas of the presented
science.We thank the Editor for the comment."Slow Down" Effect: The first, is that in your new
discussion of the "slow down" effect, you mention
that your inference is that the rate of the surface
reaction is the rate-limiting step in cement
accumulation. You then posited based on this
inference that if flow is "fast", then the reaction
"may not have time to occur". This is entirely
counter to the classical geochemical view of rate-
limiting reactions. Specifically, if the rate of the
surface reaction is the rate-limiting step, then the
overall rate of calcite growth is invariant to transport
rates. See this image that I admittedly just pulled off
of google, but it illustrates the point well.See also the response below.If calcite growth is transport limited, then growth
rates are proportional to the rates of transport
makes no difference. Thus, if you wish to continue to
argue for the "slow down" effect as differentiated
from solute sieving then that is no problem, but your
explanation of it must be consistent with the broaderPlease, refer to t                | nuanced discussion of data interpretation. There are,      | tracked in the revised manuscript.                           |  |  |  |
| addressed. I would like to reconsider this manuscript"revised manuscript version with changes tracked"
and "manuscript without tracked changes" may not
concide. The line numbering we use in this
document ( Author Response to Editor V2 ) refers to
the revised manuscript with tracked change file.Below you will find several comments related to
grammar and presentation. These are extremely
minor, as your most recent revision made major
advances in this area. The manuscript in its current
form reads very well. More major concerns are
present with respect to two areas of the presented
science.We thank the Editor for the comment."Slow Down" Effect: The first, is that in your new
discussion of the "slow down" effect, you mention
that your inference is that the rate of the surface
reaction. You then posited based on this
inference that if flow is "fast", then the reaction
"may not have time to occur". This is entirely
counter to the classical goochemical view of rate-
limiting reactions. Specifically, if the rate of the
surface reaction is the rate-limiting step, then the
overall rate of calcite growth is invariant to transport
rates. See this image that I admittedly just pulled off
of google, but it illustrates the point well.See also the response below.If calcite growth is transport limited, then growth
rates are proportional to the rates of transport
makes no difference. Thus, if you wish to continue to
argue for the "slow down" effect as differentiated
form solute sieving then that is no problem, but your
explanation of it must be consistent with the broaderSee also the response below.                                                                                                                                                                                                                                                | however, a few outstanding issues that need to be          | Please, consider that the line numbering of the              |  |  |  |
| following minor revisions. My hope is that this will
be the last round and afterwards we can proceed to
publication.and "manuscript without tracked changes" may not
coincide. The line numbering we use in this
document ( Author Response to Editor V2 ) refers to
the revised manuscript with tracked change file.Below you will find several comments related to
grammar and presentation. These are extremely
minor, as your most recent revision made major
advances in this area. The manuscript in its current
form reads very well. More major concerns are
present with respect to two areas of the presented
science.We thank the Editor for the comment."Slow Down" Effect: The first, is that in your new
discussion of the "slow down" effect, you mention
that your inference is that the rate of the surface
reaction is the rate-limiting step in cement
accumulation. You then posited based on this
inference that if flow is "fast", then the reaction
"may not have time to occur". This is entirely
counter to the classical geochemical view of rate-
limiting reactions. Specifically, if the rate of the
surface reaction limited, then present well.After some thought we decided to remove the "slow
down" effect on the presipitation from the discussion
(form solue science)If calcite growth is itransport limited, then growth
rates are proportional to the rates of transport
makes no difference. Thus, if you wish to continue to
argue for the "slow down" effect a differentiated
from solue sieving then that is no problem, but your
explanation of it must be consistent with the broader                                                                                                                                                                                                                                                                                  | addressed. I would like to reconsider this manuscript      | "revised manuscript version with changes tracked"            |  |  |  |
| be the last round and afterwards we can proceed to
publication.coincide. The line numbering we use in this
document (Author Response to Editor V2) refers to
the revised manuscript with tracked change file.Below you will find several comments related to
grammar and presentation. These are extremely
minor, as your most recent revision made major
advances in this area. The manuscript in its current
form reads very well. More major concerns are
present with respect to two areas of the presented
science.We thank the Editor for the comment."Slow Down" Effect: The first, is that in your new
discussion of the "slow down" effect, you mention
that your inference is that the rate of the surface
reaction is the rate-limiting step in cement
accumulation. You then posited based on this
inference that if flow is "fast", then the reaction
"may not have time to occur". This is entirely
counter to the classical geochemical view of rate-
limiting reactions. Specifically, if the rate of the
surface reaction limited, then growth
rates are proportional to the rates of transport
rakes no difference. Thus, if you wish to continue to
argue for the "slow down" effect as differentiated
from solute sieving then that is no problem, but your
explanation of it must be consistent with the broaderSee also the response below.                                                                                                                                                                                                                                                                                                                                                                                                                                                                                                                                                                      | following minor revisions. My hope is that this will       | and "manuscript without tracked changes" may not             |  |  |  |
| publication.document (Author Response to Editor V2) refers to
the revised manuscript with tracked change file.Below you will find several comments related to
grammar and presentation. These are extremely
minor, as your most recent revision made major
advances in this area. The manuscript in its current
form reads very well. More major concerns are
present with respect to two areas of the presented
science.We thank the Editor for the comment."Slow Down" Effect: The first, is that in your new
discussion of the "slow down" effect, you mention
that your inference is that the rate of the surface
reaction is the rate-limiting step in cement
accumulation. You then posited based on this
inference that if flow is "fast", then the reaction
"may not have time to occur". This is entirely
counter to the classical geochemical view of rate-
limiting reactions. Specifically, if the rate of the
surface reaction is the rate-limiting step, then the
overall rate of calcite growth is invariant to transport
rates. See this image that I admittedly just pulled off
of google, but it illustrates the point well.See also the response below.If calcite growth is transport limited, then growth
rates are proportional to the rates of transport.
makes no difference. Thus, if you wish to continue to
argue for the "slow down" effect as differentiated
from solute sieving then that is no problem, but your
explanation of it must be consistent with the broaderdocument (Author Response to Editor V2) refers to
the tables)If calcite growth is transportthe rate of the slow
own" effect as differentiated
from solute sieving then that is no problem, but your
explanation of it must be consistent with the broaderAfter some thought we decided to remove the "slow
down" effect as the posite.                                                                           | be the last round and afterwards we can proceed to         | coincide. The line numbering we use in this                  |  |  |  |
| the revised manuscript with tracked change file.Below you will find several comments related to
grammar and presentation. These are extremely
minor, as your most recent revision made major
advances in this area. The manuscript in its current
form reads very well. More major concerns are
present with respect to two areas of the presented
science.We thank the Editor for the comment.''Slow Down'' Effect: The first, is that in your new
discussion of the "slow down'' effect, you mention
that your inference is that the rate of the surface
reaction. You then posited based on this
inference that if flow is "fast", then the reaction
"may not have time to occur". This is entirely
counter to the classical geochemical view of rate-
limiting reactions. Specifically, if the rate of the
surface reaction limited, then the rate of transport
rates. See this image that I admittedly just pulled off
of google, but it illustrates the point well.See also the response below.If calcite growth is transport limited, then growth
rates are proportional to the rates of transport
makes no difference. Thus, if you wish to continue to
argue for the "slow down" effect as differentiated
from solute sieving then that is no problem, but your
explanation of it must be consistent with the broaderSee also the response below.                                                                                                                                                                                                                                                                                                                                                                                                                                                                                                                                                                                   | publication.                                               | document (Author Response to Editor V2) refers to            |  |  |  |
| Below you will find several comments related to
grammar and presentation. These are extremely
minor, as your most recent revision made major
advances in this area. The manuscript in its current
                                                                                                                                                                                                                                                                                                                                                                                                                                                                                                                                                                                                                                                                                                                                                                                                                                                                                                                                                                                                                                                                                                                                                                                                                                                                                                                                                                                                                                                                                                                                                                                                                                                                                                                                                                                              |                                                            | the revised manuscript with tracked change file.             |  |  |  |
| grammar and presentation. These are extremely
minor, as your most recent revision made major
advances in this area. The manuscript in its current
form reads very well. More major concerns are
present with respect to two areas of the presented
science.
"Slow Down" Effect: The first, is that in your new
discussion of the "slow down" effect, you mention
that your inference is that the rate of the surface
reaction is the rate-limiting step in cement
accumulation. You then posited based on this
inference that if flow is "fast", then the reaction
"may not have time to occur". This is entirely
counter to the classical geochemical view of rate-
limiting reactions. Specifically, if the rate of the
surface reaction is the rate-limiting step, then the
overall rate of calcite growth is invariant to transport
rates. See this image that 1 admittedly just pulled off
of google, but it illustrates the point well.
If calcite growth is transport limited, then growth
rates are proportional to the rates of transport
makes no difference. Thus, if you wish to continue to
argue for the "slow down" effect as differentiated
from solute sieving then that is no problem, but your
explanation of it must be consistent with the broader                                                                                                                                                                                                                                                                                                                                                                                                                                                                                                                                                                                                                                                             | Below you will find several comments related to            | We thank the Editor for the comment.                         |  |  |  |
|  <li>minor, as your most recent revision made major advances in this area. The manuscript in its current form reads very well. More major concerns are present with respect to two areas of the presented science.</li> <li>"Slow Down" Effect: The first, is that in your new discussion of the "slow down" effect, you mention that your inference is that the rate of the surface reaction is the rate-limiting step in cement accumulation. You then posited based on this inference that if flow is "fast", then the reaction "may not have time to occur". This is entirely counter to the classical geochemical view of rate-limiting reactions. Specifically, if the rate of the surface reaction is the rate-limiting step, then the overall rate of calcite growth is invariant to transport rates. See this image that I admittedly just pulled off of google, but it illustrates the point well.</li> <li>If calcite growth is transport limited, then growth rates are proportional to the rates of transport. If it is surface reaction limited, then the rate of transport makes no difference. Thus, if you wish to continue to argue for the "slow down" effect as differentiated from solute sieving then that is no problem, but your explanation of it must be consistent with the broader</li>                                                                                                                                                                                                                                                                                                                                                                                                                                                                                                                                                                                                                                                                                | grammar and presentation. These are extremely              |                                                              |  |  |  |
| advances in this area. The manuscript in its current
form reads very well. More major concerns are
present with respect to two areas of the presented
science.
"Slow Down" Effect: The first, is that in your new
discussion of the "slow down" effect, you mention
that your inference is that the rate of the surface
reaction is the rate-limiting step in cement
accumulation. You then posited based on this
inference that if flow is "fast", then the reaction
"may not have time to occur". This is entirely
counter to the classical geochemical view of rate-
limiting reactions. Specifically, if the rate of the
surface reaction is the rate-limiting step, then the
overall rate of calcite growth is invariant to transport
rates. See this image that I admittedly just pulled off
of google, but it illustrates the point well.
If calcite growth is transport limited, then growth
rates are proportional to the rates of transport. If it is
surface reaction limited, then the rate of transport
makes no difference. Thus, if you wish to continue to
argue for the "slow down" effect as differentiated
from solute sieving then that is no problem, but your
explanation of it must be consistent with the broader                                                                                                                                                                                                                                                                                                                                                                                                                                                                                                                                                                                                                                                                                              | minor, as your most recent revision made major             | Please, refer to the following responses for detailed        |  |  |  |
| form reads very well. More major concerns are
present with respect to two areas of the presented
science.of the Editor."Slow Down" Effect: The first, is that in your new
discussion of the "slow down" effect, you mention
that your inference is that the rate of the surface
reaction is the rate-limiting step in cement
accumulation. You then posited based on this
inference that if flow is "fast", then the reaction
"may not have time to occur". This is entirely
counter to the classical geochemical view of rate-
limiting reactions. Specifically, if the rate of the
surface reaction is the rate-limiting step, then the
overall rate of calcite growth is invariant to transport
rates. See this image that I admittedly just pulled off
of google, but it illustrates the point well.See also the response below.If calcite growth is transport limited, then growth
rates are proportional to the rates of transport
makes no difference. Thus, if you wish to continue to
argue for the "slow down" effect as differentiated
from solute sieving then that is no problem, but your
explanation of it must be consistent with the broaderof the Editor.                                                                                                                                                                                                                                                                                                                                                                                                                                                                                                                                                                                                                                                                                                                                                                        | advances in this area. The manuscript in its current       | explanations and revisions made about the comments           |  |  |  |
| present with respect to two areas of the presented
science.After some thought we decided to remove the "slow
down" effect: The first, is that in your new
discussion of the "slow down" effect, you mention
that your inference is that the rate of the surface
reaction is the rate-limiting step in cement
accumulation. You then posited based on this
inference that if flow is "fast", then the reaction
"may not have time to occur". This is entirely
counter to the classical geochemical view of rate-
limiting reaction. Specifically, if the rate of the
surface reaction is the rate-limiting step, then the
overall rate of calcite growth is invariant to transport
rates. See this image that I admittedly just pulled off
of google, but it illustrates the point well.After some thought we decided to remove the "slow
down" effect on the precipitation from the discussion
(former lines 576-577) since we have not yet a solid
physico-chemical basis to support it. Now we keep it
as an observation (lines 566-574).If calcite growth is invariant to transport
rates are proportional to the rates of transport.
If it is
surface reaction limited, then growth
rates are proportional to the rates of transport
makes no difference. Thus, if you wish to continue to
argue for the "slow down" effect as differentiated
from solute sieving then that is no problem, but your
explanation of it must be consistent with the broaderAfter some thought we decided to remove the "slow
down" effect on the precipitation from the discussion
(former lines 576-577) since we have not yet a solid
physico-chemical basis to support it. Now we keep it
as an observation (lines 566-574).                                                                                                                                                                                                 | form reads very well. More major concerns are              | of the Editor.                                               |  |  |  |
| science."Slow Down" Effect: The first, is that in your new
discussion of the "slow down" effect, you mention
that your inference is that the rate of the surface
reaction is the rate-limiting step in cement
accumulation. You then posited based on this
inference that if flow is "fast", then the reaction
"may not have time to occur". This is entirely
counter to the classical geochemical view of rate-
limiting reactions. Specifically, if the rate of the
surface reaction is the rate-limiting step, then the
overall rate of calcite growth is invariant to transport
rates. See this image that I admittedly just pulled off
of google, but it illustrates the point well.See also the response below.If calcite growth is transport limited, then growth
rates are proportional to the rates of transport
makes no difference. Thus, if you wish to continue to
argue for the "slow down" effect as differentiated
from solute sieving then that is no problem, but your
explanation of it must be consistent with the broaderAfter some thought we decided to remove the "slow
down" effect on the precipitation from the discussion
(former lines 576-577) since we have not yet a solid
physico-chemical basis to support it. Now we keep it
as an observation (lines 566-574).See also the response below.See also the response below.                                                                                                                                                                                                                                                                                                                                                                                                                                                                                                                                                                                   | present with respect to two areas of the presented         |                                                              |  |  |  |
|  <li>"Slow Down" Effect: The first, is that in your new discussion of the "slow down" effect, you mention that your inference is that the rate of the surface reaction is the rate-limiting step in cement accumulation. You then posited based on this inference that if flow is "fast", then the reaction "may not have time to occur". This is entirely counter to the classical geochemical view of rate-limiting reactions. Specifically, if the rate of the surface reaction is the rate-limiting step, then the overall rate of calcite growth is invariant to transport rates. See this image that I admittedly just pulled off of google, but it illustrates the point well.</li> <li>If calcite growth is transport limited, then growth rates are proportional to the rates of transport. If it is surface reaction limited, then the rate of transport makes no difference. Thus, if you wish to continue to argue for the "slow down" effect as differentiated from solute sieving then that is no problem, but your explanation of it must be consistent with the broader</li>                                                                                                                                                                                                                                                                                                                                                                                                                                                                                                                                                                                                                                                                                                                                                                                                                                                                                                       | science.                                                   |                                                              |  |  |  |
| discussion of the "slow down" effect, you mention
that your inference is that the rate of the surface
reaction is the rate-limiting step in cement
accumulation. You then posited based on this
inference that if flow is "fast", then the reaction
"may not have time to occur". This is entirely
counter to the classical geochemical view of rate-
limiting reactions. Specifically, if the rate of the
surface reaction is the rate-limiting step, then the
overall rate of calcite growth is invariant to transport
rates. See this image that I admittedly just pulled off
of google, but it illustrates the point well.
If calcite growth is transport limited, then growth
rates are proportional to the rates of transport. If it is
surface reaction limited, then the rate of transport
makes no difference. Thus, if you wish to continue to
argue for the "slow down" effect as differentiated
from solute sieving then that is no problem, but your
explanation of it must be consistent with the broader                                                                                                                                                                                                                                                                                                                                                                                                                                                                                                                                                                                                                                                                                                                                                                                                                                                                                                                               | "Slow Down" Effect: The first, is that in your new         | After some thought we decided to remove the "slow            |  |  |  |
|  <li>that your inference is that the rate of the surface reaction is the rate-limiting step in cement accumulation. You then posited based on this inference that if flow is "fast", then the reaction "may not have time to occur". This is entirely counter to the classical geochemical view of rate-limiting reactions. Specifically, if the rate of the surface reaction is the rate-limiting step, then the overall rate of calcite growth is invariant to transport rates. See this image that I admittedly just pulled off of google, but it illustrates the point well.</li> <li>If calcite growth is transport limited, then growth rates are proportional to the rates of transport. If it is surface reaction limited, then the rate of transport makes no difference. Thus, if you wish to continue to argue for the "slow down" effect as differentiated from solute sieving then that is no problem, but your explanation of it must be consistent with the broader</li>                                                                                                                                                                                                                                                                                                                                                                                                                                                                                                                                                                                                                                                                                                                                                                                                                                                                                                                                                                                                            | discussion of the "slow down" effect, you mention          | down" effect on the precipitation from the discussion        |  |  |  |
| reaction is the rate-limiting step in cement
accumulation. You then posited based on this
inference that if flow is "fast", then the reaction
"may not have time to occur". This is entirely
counter to the classical geochemical view of rate-
limiting reactions. Specifically, if the rate of the
surface reaction is the rate-limiting step, then the
overall rate of calcite growth is invariant to transport
rates. See this image that I admittedly just pulled off
of google, but it illustrates the point well.
If calcite growth is transport limited, then growth
rates are proportional to the rates of transport. If it is
surface reaction limited, then the rate of transport
makes no difference. Thus, if you wish to continue to
argue for the "slow down" effect as differentiated
from solute sieving then that is no problem, but your
explanation of it must be consistent with the broader                                                                                                                                                                                                                                                                                                                                                                                                                                                                                                                                                                                                                                                                                                                                                                                                                                                                                                                                                                                                                                           | that your inference is that the rate of the surface        | (former lines 576-577) since we have not yet a solid         |  |  |  |
| accumulation. You then posited based on this
inference that if flow is "fast", then the reaction
"may not have time to occur". This is entirely
counter to the classical geochemical view of rate-
limiting reactions. Specifically, if the rate of the
surface reaction is the rate-limiting step, then the
overall rate of calcite growth is invariant to transport
rates. See this image that I admittedly just pulled off
of google, but it illustrates the point well.
If calcite growth is transport limited, then growth
rates are proportional to the rates of transport. If it is
surface reaction limited, then the rate of transport
makes no difference. Thus, if you wish to continue to
argue for the "slow down" effect as differentiated
from solute sieving then that is no problem, but your
explanation of it must be consistent with the broader                                                                                                                                                                                                                                                                                                                                                                                                                                                                                                                                                                                                                                                                                                                                                                                                                                                                                                                                                                                                                                                                                           | reaction is the rate-limiting step in cement               | physico-chemical basis to support it. Now we keep it         |  |  |  |
|  <li>inference that if flow is "fast", then the reaction</li> <li>"may not have time to occur". This is entirely</li> <li>counter to the classical geochemical view of rate-</li> <li>limiting reactions. Specifically, if the rate of the</li> <li>surface reaction is the rate-limiting step, then the</li> <li>overall rate of calcite growth is invariant to transport</li> <li>rates. See this image that I admittedly just pulled off</li> <li>of google, but it illustrates the point well.</li> <li>If calcite growth is transport limited, then growth</li> <li>rates are proportional to the rates of transport. If it is</li> <li>surface reaction limited, then the rate of transport</li> <li>makes no difference. Thus, if you wish to continue to</li> <li>argue for the "slow down" effect as differentiated</li> <li>from solute sieving then that is no problem, but your</li> <li>explanation of it must be consistent with the broader</li>                                                                                                                                                                                                                                                                                                                                                                                                                                                                                                                                                                                                                                                                                                                                                                                                                                                                                                                                                                                                                                    | accumulation. You then posited based on this               | as an observation (lines 566-574).                           |  |  |  |
|  <li>"may not have time to occur". This is entirely counter to the classical geochemical view of rate-limiting reactions. Specifically, if the rate of the surface reaction is the rate-limiting step, then the overall rate of calcite growth is invariant to transport rates. See this image that I admittedly just pulled off of google, but it illustrates the point well.</li> <li>If calcite growth is transport limited, then growth rates are proportional to the rate of transport. If it is surface reaction limited, then the rate of transport makes no difference. Thus, if you wish to continue to argue for the "slow down" effect as differentiated from solute sieving then that is no problem, but your explanation of it must be consistent with the broader</li>                                                                                                                                                                                                                                                                                                                                                                                                                                                                                                                                                                                                                                                                                                                                                                                                                                                                                                                                                                                                                                                                                                                                                                                                               | inference that if flow is "fast", then the reaction        |                                                              |  |  |  |
| counter to the classical geochemical view of rate-
limiting reactions. Specifically, if the rate of the
surface reaction is the rate-limiting step, then the
overall rate of calcite growth is invariant to transport
rates. See this image that I admittedly just pulled off
of google, but it illustrates the point well.
If calcite growth is transport limited, then growth
rates are proportional to the rates of transport. If it is
surface reaction limited, then the rate of transport
makes no difference. Thus, if you wish to continue to
argue for the "slow down" effect as differentiated
from solute sieving then that is no problem, but your
explanation of it must be consistent with the broader                                                                                                                                                                                                                                                                                                                                                                                                                                                                                                                                                                                                                                                                                                                                                                                                                                                                                                                                                                                                                                                                                                                                                                                                                                                    | "may not have time to occur". This is entirely             |                                                              |  |  |  |
|  <li>limiting reactions. Specifically, if the rate of the surface reaction is the rate-limiting step, then the overall rate of calcite growth is invariant to transport rates. See this image that I admittedly just pulled off of google, but it illustrates the point well.</li> <li>If calcite growth is transport limited, then growth rates are proportional to the rates of transport. If it is surface reaction limited, then the rate of transport makes no difference. Thus, if you wish to continue to argue for the "slow down" effect as differentiated from solute sieving then that is no problem, but your explanation of it must be consistent with the broader</li>                                                                                                                                                                                                                                                                                                                                                                                                                                                                                                                                                                                                                                                                                                                                                                                                                                                                                                                                                                                                                                                                                                                                                                                                                                                                                                               | counter to the classical geochemical view of rate-         | See also the response below.                                 |  |  |  |
| surface reaction is the rate-limiting step, then the
overall rate of calcite growth is invariant to transport
rates. See this image that I admittedly just pulled off
of google, but it illustrates the point well.
If calcite growth is transport limited, then growth
rates are proportional to the rates of transport. If it is
surface reaction limited, then the rate of transport
makes no difference. Thus, if you wish to continue to
argue for the "slow down" effect as differentiated
from solute sieving then that is no problem, but your
explanation of it must be consistent with the broader                                                                                                                                                                                                                                                                                                                                                                                                                                                                                                                                                                                                                                                                                                                                                                                                                                                                                                                                                                                                                                                                                                                                                                                                                                                                                                                                                                  | limiting reactions. Specifically, if the rate of the       | Å                                                            |  |  |  |
| overall rate of calcite growth is invariant to transport
rates. See this image that I admittedly just pulled off
of google, but it illustrates the point well.
If calcite growth is transport limited, then growth
rates are proportional to the rates of transport. If it is
surface reaction limited, then the rate of transport
makes no difference. Thus, if you wish to continue to
argue for the "slow down" effect as differentiated
from solute sieving then that is no problem, but your
explanation of it must be consistent with the broader                                                                                                                                                                                                                                                                                                                                                                                                                                                                                                                                                                                                                                                                                                                                                                                                                                                                                                                                                                                                                                                                                                                                                                                                                                                                                                                                                                                                                          | surface reaction is the rate-limiting step, then the       |                                                              |  |  |  |
| rates. See this image that I admittedly just pulled off
of google, but it illustrates the point well.
If calcite growth is transport limited, then growth
rates are proportional to the rates of transport. If it is
surface reaction limited, then the rate of transport
makes no difference. Thus, if you wish to continue to
argue for the "slow down" effect as differentiated
from solute sieving then that is no problem, but your
explanation of it must be consistent with the broader                                                                                                                                                                                                                                                                                                                                                                                                                                                                                                                                                                                                                                                                                                                                                                                                                                                                                                                                                                                                                                                                                                                                                                                                                                                                                                                                                                                                                                                                                      | overall rate of calcite growth is invariant to transport   |                                                              |  |  |  |
| of google, but it illustrates the point well.
If calcite growth is transport limited, then growth
rates are proportional to the rates of transport. If it is
surface reaction limited, then the rate of transport
makes no difference. Thus, if you wish to continue to
argue for the "slow down" effect as differentiated
from solute sieving then that is no problem, but your
explanation of it must be consistent with the broader                                                                                                                                                                                                                                                                                                                                                                                                                                                                                                                                                                                                                                                                                                                                                                                                                                                                                                                                                                                                                                                                                                                                                                                                                                                                                                                                                                                                                                                                                                                                                 | rates. See this image that I admittedly just pulled off    |                                                              |  |  |  |
| If calcite growth is transport limited, then growth
rates are proportional to the rates of transport. If it is
surface reaction limited, then the rate of transport
makes no difference. Thus, if you wish to continue to
argue for the "slow down" effect as differentiated
from solute sieving then that is no problem, but your
explanation of it must be consistent with the broader                                                                                                                                                                                                                                                                                                                                                                                                                                                                                                                                                                                                                                                                                                                                                                                                                                                                                                                                                                                                                                                                                                                                                                                                                                                                                                                                                                                                                                                                                                                                                                                                  | of google, but it illustrates the point well.              |                                                              |  |  |  |
| If calcite growth is transport limited, then growth
rates are proportional to the rates of transport. If it is
surface reaction limited, then the rate of transport
makes no difference. Thus, if you wish to continue to
argue for the "slow down" effect as differentiated
from solute sieving then that is no problem, but your
explanation of it must be consistent with the broader                                                                                                                                                                                                                                                                                                                                                                                                                                                                                                                                                                                                                                                                                                                                                                                                                                                                                                                                                                                                                                                                                                                                                                                                                                                                                                                                                                                                                                                                                                                                                                                                  |                                                            |                                                              |  |  |  |
| rates are proportional to the rates of transport. If it is
surface reaction limited, then the rate of transport
makes no difference. Thus, if you wish to continue to
argue for the "slow down" effect as differentiated
from solute sieving then that is no problem, but your
explanation of it must be consistent with the broader                                                                                                                                                                                                                                                                                                                                                                                                                                                                                                                                                                                                                                                                                                                                                                                                                                                                                                                                                                                                                                                                                                                                                                                                                                                                                                                                                                                                                                                                                                                                                                                                                                                         | If calcite growth is transport limited then growth         |                                                              |  |  |  |
| surface reaction limited, then the rate of transport
makes no difference. Thus, if you wish to continue to
argue for the "slow down" effect as differentiated
from solute sieving then that is no problem, but your
explanation of it must be consistent with the broader                                                                                                                                                                                                                                                                                                                                                                                                                                                                                                                                                                                                                                                                                                                                                                                                                                                                                                                                                                                                                                                                                                                                                                                                                                                                                                                                                                                                                                                                                                                                                                                                                                                                                                                       | rates are proportional to the rates of transport. If it is |                                                              |  |  |  |
| makes no difference. Thus, if you wish to continue to
argue for the "slow down" effect as differentiated
from solute sieving then that is no problem, but your
explanation of it must be consistent with the broader                                                                                                                                                                                                                                                                                                                                                                                                                                                                                                                                                                                                                                                                                                                                                                                                                                                                                                                                                                                                                                                                                                                                                                                                                                                                                                                                                                                                                                                                                                                                                                                                                                                                                                                                                                               | surface reaction limited then the rate of transport        |                                                              |  |  |  |
| argue for the "slow down" effect as differentiated
from solute sieving then that is no problem, but your
explanation of it must be consistent with the broader                                                                                                                                                                                                                                                                                                                                                                                                                                                                                                                                                                                                                                                                                                                                                                                                                                                                                                                                                                                                                                                                                                                                                                                                                                                                                                                                                                                                                                                                                                                                                                                                                                                                                                                                                                                                                                        | makes no difference. Thus, if you wish to continue to      |                                                              |  |  |  |
| from solute sieving then that is no problem, but your
explanation of it must be consistent with the broader                                                                                                                                                                                                                                                                                                                                                                                                                                                                                                                                                                                                                                                                                                                                                                                                                                                                                                                                                                                                                                                                                                                                                                                                                                                                                                                                                                                                                                                                                                                                                                                                                                                                                                                                                                                                                                                                                              | argue for the "slow down" effect as differentiated         |                                                              |  |  |  |
| explanation of it must be consistent with the broader                                                                                                                                                                                                                                                                                                                                                                                                                                                                                                                                                                                                                                                                                                                                                                                                                                                                                                                                                                                                                                                                                                                                                                                                                                                                                                                                                                                                                                                                                                                                                                                                                                                                                                                                                                                                                                                                                                                                                       | from solute sieving then that is no problem, but your      |                                                              |  |  |  |
| explanation of thirds be consistent with the broader                                                                                                                                                                                                                                                                                                                                                                                                                                                                                                                                                                                                                                                                                                                                                                                                                                                                                                                                                                                                                                                                                                                                                                                                                                                                                                                                                                                                                                                                                                                                                                                                                                                                                                                                                                                                                                                                                                                                                        | explanation of it must be consistent with the broader      |                                                              |  |  |  |
| hody of knowledge considering reactive transport. In                                                                                                                                                                                                                                                                                                                                                                                                                                                                                                                                                                                                                                                                                                                                                                                                                                                                                                                                                                                                                                                                                                                                                                                                                                                                                                                                                                                                                                                                                                                                                                                                                                                                                                                                                                                                                                                                                                                                                        | hody of knowledge considering reactive transport. In       |                                                              |  |  |  |
| short. I think your proposed mechanism needs a hit                                                                                                                                                                                                                                                                                                                                                                                                                                                                                                                                                                                                                                                                                                                                                                                                                                                                                                                                                                                                                                                                                                                                                                                                                                                                                                                                                                                                                                                                                                                                                                                                                                                                                                                                                                                                                                                                                                                                                          | short I think your proposed mechanism needs a bit          |                                                              |  |  |  |
| more thought                                                                                                                                                                                                                                                                                                                                                                                                                                                                                                                                                                                                                                                                                                                                                                                                                                                                                                                                                                                                                                                                                                                                                                                                                                                                                                                                                                                                                                                                                                                                                                                                                                                                                                                                                                                                                                                                                                                                                                                                | more thought                                               |                                                              |  |  |  |
| Oxygen Isotope Interpretations: Lappreciate your We have stated the fluid $\delta^{18}$ O values in the                                                                                                                                                                                                                                                                                                                                                                                                                                                                                                                                                                                                                                                                                                                                                                                                                                                                                                                                                                                                                                                                                                                                                                                                                                                                                                                                                                                                                                                                                                                                                                                                                                                                                                                                                                                                                                                                                                     | Oxygen Isotope Interpretations: Lappreciate your           | We have stated the fluid $\delta^{18}\Omega$ values in the   |  |  |  |
| doing the calculation to estimate fluid d180 values.                                                                                                                                                                                                                                                                                                                                                                                                                                                                                                                                                                                                                                                                                                                                                                                                                                                                                                                                                                                                                                                                                                                                                                                                                                                                                                                                                                                                                                                                                                                                                                                                                                                                                                                                                                                                                                                                                                                                                        | doing the calculation to estimate fluid d180 values        | manuscript and we have added a brief discussion              |  |  |  |
| but in this version you simply mention that you did considering the $\delta^{18}$ O values expected for meteoric                                                                                                                                                                                                                                                                                                                                                                                                                                                                                                                                                                                                                                                                                                                                                                                                                                                                                                                                                                                                                                                                                                                                                                                                                                                                                                                                                                                                                                                                                                                                                                                                                                                                                                                                                                                                                                                                                            | but in this version you simply mention that you did        | considering the $\delta^{18}$ O values expected for meteoric |  |  |  |

| that and that they are consistent with your
interpretations. The values need to be: 1) stated
explicitly in the manuscript (rather than pointing the
reader toward the Supplement); and 2) discussed
relative to the expected values for meteoric fluids in
the study area. The values you list in the supplement
are a bit heavy for meteoric fluids - is this consistent
with what has been observed for modern
precipitation/ground water in the study area, or better
yet, any values obtained from paleoclimatic studies?                                                                                               | water for both Loiano (lines 695-702) and Bollène
(lines 753-756 and 766-768).
Unfortunately, we do not have any direct constrains
on the timing of cement precipitation, thus comparing
the fluid $\delta^{18}$ O we found with those from the past
could be challenging, and they might also not be
available for the area.
However, we are considering performing some U-Pb
dating of cements in the future, to constrain the time
of cement formation. This would allow us also to
better infer the depth at which the precipitation took
place. We have also added a line to highlight that
further analysis are necessary to better constrain the
conditions of the cementing fluids (lines 702-703). |
|---------------------------------------------------------------------------------------------------------------------------------------------------------------------------------------------------------------------------------------------------------------------------------------------------------------------------------------------------------------------------------------------------------------------------------------------------------------------------------------------------------------------------------------------------------------------------------------------------------------------------------------------------------|----------------------------------------------------------------------------------------------------------------------------------------------------------------------------------------------------------------------------------------------------------------------------------------------------------------------------------------------------------------------------------------------------------------------------------------------------------------------------------------------------------------------------------------------------------------------------------------------------------------------------------------------------------------------------------------------------------------------------------------------------|
| See: Giustini, Francesca, Mauro Brilli, and Antonio
Patera. "Mapping oxygen stable isotopes of
precipitation in Italy." Journal of Hydrology:
Regional Studies 8 (2016): 162-181                                                                                                                                                                                                                                                                                                                                                                                                                                                               | The reference was added (lines 1041-1042) and used
in the text (lines 697-698) to discuss the $\delta^{18}O_{\text{fluid}}$ we
calculated with those expected for meteoric fluids in
the study area. See also the response just above                                                                                                                                                                                                                                                                                                                                                                                                                                                                                                     |
| Specific Comments                                                                                                                                                                                                                                                                                                                                                                                                                                                                                                                                                                                                                                       | the study from see also the response just above.                                                                                                                                                                                                                                                                                                                                                                                                                                                                                                                                                                                                                                                                                                   |
| 89: As this sentence is currently constructed, an additional comma after "between" would be preferable.                                                                                                                                                                                                                                                                                                                                                                                                                                                                                                                                                 | Ok, we have done that (line 89).                                                                                                                                                                                                                                                                                                                                                                                                                                                                                                                                                                                                                                                                                                                   |
| 92: Suggest rephrasing for english grammar clarity:
"Our study also allows evaluation of the impact of"                                                                                                                                                                                                                                                                                                                                                                                                                                                                                                                                              | Ok, suggestion taken and implemented (line 92).                                                                                                                                                                                                                                                                                                                                                                                                                                                                                                                                                                                                                                                                                                    |
| 252: Thank you for adopting the more standard azimuth notation. Something has gone wrong here though - N334E?? That does not make sense. It should just be left at "334°". The N and E information are redundant. Please address here and throughout.                                                                                                                                                                                                                                                                                                                                                                                                   | Ok, we have done that (lines 189 and 261-262).                                                                                                                                                                                                                                                                                                                                                                                                                                                                                                                                                                                                                                                                                                     |
| 544-546: There is a problem with the logic here, in that a classical view of surface reaction limited growth implies that the overall growth rate is invariant with respect to the rate of transport.                                                                                                                                                                                                                                                                                                                                                                                                                                                   | The Editor is right. The sentence was removed
(former lines 576-577) as also the discussion about
the "slow down" effect on the precipitation. Please,
see also the response above in the major comments
section.                                                                                                                                                                                                                                                                                                                                                                                                                                                                                                                      |
| 555-560: The point I was making in my previous
comments was that we very frequently see
preferential calcite cementation of deformation bands
that do not record significant cataclasis or fracturing.
This implies that there must be something else (or at
least something in addition to the role of fracturing)
in helping to localize calcite cements in those
structures. This makes them different from quartz
cements, where the main thing seems to be
fracturing. You do not need to change anything here.
I bring this up only as an interesting point of
discussion and potential impetus for future work. | Yes, this is an interesting point indeed.
We thank the Editor for the comment.                                                                                                                                                                                                                                                                                                                                                                                                                                                                                                                                                                                                                                                                  |
| 585-590: I appreciate your adding this paragraph in
response to my comments, but I do not think you
actually need it. The other modifications you have
made to this section make it a bit superfluous. You
are welcome to keep it if you chose, but its                                                                                                                                                                                                                                                                                                                                                                                     | we nave removed that paragraph (former lines 645-646).                                                                                                                                                                                                                                                                                                                                                                                                                                                                                                                                                                                                                                                                                             |

| elimination may make for more concise reading.                                                                                                                                                                                                                                                                                                                                                                                                                                                                                                                                                                                                                                                                                                                                             |                                                                                                                                                                                                                                                                                                                                                                                                                                                                                                                                                                                                                                                                          |
|--------------------------------------------------------------------------------------------------------------------------------------------------------------------------------------------------------------------------------------------------------------------------------------------------------------------------------------------------------------------------------------------------------------------------------------------------------------------------------------------------------------------------------------------------------------------------------------------------------------------------------------------------------------------------------------------------------------------------------------------------------------------------------------------|--------------------------------------------------------------------------------------------------------------------------------------------------------------------------------------------------------------------------------------------------------------------------------------------------------------------------------------------------------------------------------------------------------------------------------------------------------------------------------------------------------------------------------------------------------------------------------------------------------------------------------------------------------------------------|
| 611-621: This still needs work. The opening                                                                                                                                                                                                                                                                                                                                                                                                                                                                                                                                                                                                                                                                                                                                                | We revised the paragraph considering the Editor                                                                                                                                                                                                                                                                                                                                                                                                                                                                                                                                                                                                                          |
| sentence should more reasonably stated that                                                                                                                                                                                                                                                                                                                                                                                                                                                                                                                                                                                                                                                                                                                                                | comment. We nope that now it is clearer (lines 692-                                                                                                                                                                                                                                                                                                                                                                                                                                                                                                                                                                                                                      |
| Oxygen isotope data also support a meteoric                                                                                                                                                                                                                                                                                                                                                                                                                                                                                                                                                                                                                                                                                                                                                | 705).                                                                                                                                                                                                                                                                                                                                                                                                                                                                                                                                                                                                                                                                    |
| environment" or similar, and then go on to state the                                                                                                                                                                                                                                                                                                                                                                                                                                                                                                                                                                                                                                                                                                                                       |                                                                                                                                                                                                                                                                                                                                                                                                                                                                                                                                                                                                                                                                          |
| nothing on their own, and I also find the notion of
any particular d13C value being indicative of
meteoric environments preferentially questionable.                                                                                                                                                                                                                                                                                                                                                                                                                                                                                                                                                                                                                                 | manuscript and we have added a brief discussion
considering the $\delta^{18}$ O values expected for present-time
meteoric fluids characterizing the area of Loiano
(lines 605, 702)                                                                                                                                                                                                                                                                                                                                                                                                                                                                             |
| that are consistent with a meteoric origin, but I do                                                                                                                                                                                                                                                                                                                                                                                                                                                                                                                                                                                                                                                                                                                                       | (Intes 093-702).                                                                                                                                                                                                                                                                                                                                                                                                                                                                                                                                                                                                                                                         |
|  <li>not think they can be an independent indicator.</li> <li>Oxygen, however, can. Toward that end, a bigger problem is that you mention calculating the fluid d18O, but do not discuss the inferred values here? Its a fundamental data type you argue to support your interpretation - it cannot be relegated to the supplemental information and not stated. The calculation can reasonably reside in the supplement, but you need to actually state the inferred fluid d18O values.</li> <li>I see in the supplement that the values you calculate are between about -7 and -3. These actually strike me as slightly heavy for meteoric fluids, but those do vary by quite a lot over the surface of the earth. You will also need some brief discussion as to whether</li>  | Unfortunately, we do not have any direct constrains
on the timing of cement precipitation, thus comparing
the fluid $\delta^{18}$ O we found with those from
paleoclimatic the past could be challenging.
However, we are considering performing some U-Pb
dating of cements in the future, to constrain the time
of cement formation. This would allow us also to
better infer the depth at which the precipitation took
place. We have also added a line to highlight what a
future research direction could be to properly
constrain the conditions of the cementing fluids (lines
702-703).
Please, see also the response above. |
| these values are consistent with meteoric fluids in the                                                                                                                                                                                                                                                                                                                                                                                                                                                                                                                                                                                                                                                                                                                                    |                                                                                                                                                                                                                                                                                                                                                                                                                                                                                                                                                                                                                                                                          |
| study areas, either now or from a paleoclimatic                                                                                                                                                                                                                                                                                                                                                                                                                                                                                                                                                                                                                                                                                                                                            |                                                                                                                                                                                                                                                                                                                                                                                                                                                                                                                                                                                                                                                                          |
| perspective around the inferred time of cement                                                                                                                                                                                                                                                                                                                                                                                                                                                                                                                                                                                                                                                                                                                                             |                                                                                                                                                                                                                                                                                                                                                                                                                                                                                                                                                                                                                                                                          |
| formation.                                                                                                                                                                                                                                                                                                                                                                                                                                                                                                                                                                                                                                                                                                                                                                                 | Ob the contenes was shereed (line 724 725)                                                                                                                                                                                                                                                                                                                                                                                                                                                                                                                                                                                                                               |
| where fluids slow down in the vicinity of DBs.                                                                                                                                                                                                                                                                                                                                                                                                                                                                                                                                                                                                                                                                                                                                             | Ok, the sentence was changed (line 754-755).                                                                                                                                                                                                                                                                                                                                                                                                                                                                                                                                                                                                                             |
| 650-655: Same problem mentioned above.                                                                                                                                                                                                                                                                                                                                                                                                                                                                                                                                                                                                                                                                                                                                                     | We revised the paragraph considering the Editor
comment. We hope that now it is clearer (lines 751-
768).                                                                                                                                                                                                                                                                                                                                                                                                                                                                                                                                                          |
|                                                                                                                                                                                                                                                                                                                                                                                                                                                                                                                                                                                                                                                                                                                                                                                            | We have stated the fluid $\delta^{18}$ O values in the manuscript and we have added a brief discussion considering the $\delta^{18}$ O values expected for present-time metaoric fluids characterizing the area of Ballàna                                                                                                                                                                                                                                                                                                                                                                                                                                               |
|                                                                                                                                                                                                                                                                                                                                                                                                                                                                                                                                                                                                                                                                                                                                                                                            | (lines 753-756 and 766-768).                                                                                                                                                                                                                                                                                                                                                                                                                                                                                                                                                                                                                                             |

[revised manuscript text omitted]